## RESEARCH ARTICLE

# Birthdate aligns vestibular sensory neurons with central and motor partners across a sensorimotor reflex circuit for gaze stabilization

Stephanie Huang[1,2], Emily Gershowitz[1], Marie R. Greaney[1], Samantha N. Davis[1], David Schoppik[1] and Dena Goldblatt[1,2,3,*]

## ABSTRACT

Developing populations of connected neurons often share spatial and/or temporal features that anticipate their assembly. A unifying spatiotemporal motif might link sensory, central and motor populations that comprise an entire circuit. In the sensorimotor reflex circuit that stabilizes vertebrate gaze, central and motor partners are paired in time (birthdate) and space (dorsoventral). To determine if birthdate and/or dorsoventral organization could align the entire circuit, we measured the spatial and temporal development of the sensory circuit node: the vestibular ganglion neurons. We discovered that progressive dorsal-to-ventral organization closely predicts vestibular ganglion development, with additional organization along its functional (rostrocaudal) axis. With an acute optical lesion and calcium imaging paradigm, we found that this common temporal axis anticipated functional sensory-to-central partner matching. We propose a 'first-come, first-served' model, in which birthdate organizes and assembles the sensory, central and motor populations that comprise the gaze stabilization circuit, a general strategy for polysynaptic circuit assembly across embryonically diverse neural populations.

KEY WORDS: Zebrafish, Development, Vestibular, Sensory, Neural circuits, Neuroscience

## INTRODUCTION

Individual populations that comprise neural circuits develop in both space and time. During development and maturation, spatial relationships among somata and/or neuropil impose structural constraints on neural circuit assembly and function (Balaskas et al., 2020; Bikoff et al., 2016; Cang and Feldheim, 2013; Knudsen and Brainard, 1991; Sürmeli et al., 2011), leaving a topographic record (Erzurumlu et al., 2010; Levine et al., 2012; Penfield and Boldrey, 1937). Progressive neurogenesis can generate similar maps in time, instructing synaptic partner matching across paired populations (Fetcho and McLean, 2010; Hamid et al., 2024; Holguera and Desplan, 2018; Kishore et al., 2020; Mark et al., 2021; Pujol-Martí

[1]Departments of Neuroscience, Otolaryngology, and the Institute for Translational Neuroscience, NYU Grossman School of Medicine, New York, NY 10016, USA. [2]Center for Neural Science, New York University, New York, NY 10003, USA. [3]Spinal Circuits and Plasticity Unit, National Institute of Neurological Disorders and Stroke, Bethesda, MD 20814, USA.

*Author for correspondence (goldblatt.dena@gmail.com)

S.H., 0009-0002-0609-1655; E.G., 0009-0008-4732-4340; M.R.G., 0000-0001-9552-7566; S.N.D., 0000-0003-2704-9480; D.S., 0000-0001-7969-9632; D.G., 0000-0003-0333-2433

et al., 2012; McLean and Fetcho, 2009; Meng et al., 2019; Pujala and Koyama, 2019; Seroka and Doe, 2019; Wreden et al., 2017). While the alignment of individual partner populations in space and time is well-established, sensorimotor circuits are almost all polysynaptic. Further, sensorimotor circuits generally span embryologically-diverse tissue, opening the question of whether the spatial and temporal features that pair partners might similarly organize an entire circuit.

The vertebrate vestibulo-ocular reflex is a canonical sensorimotor circuit that uses precise sensory-to-motor partner matching to stabilize gaze. The circuit transforms two vertical directions of head/body destabilization (nose-up/nose-down) into compensatory eye rotation behavior (eyes-down/eyes-up). Anatomically, the circuit consists of hair cells in the inner ear, sensory afferents in the anterior (vestibular) statoacoustic ganglion, central vestibular projection neurons, motor neurons and extraocular muscles (Szentágothai, 1964). Each direction of motion (up/down) is segregated into its own specific channel comprised of directionally-selective hair cells, neurons and muscles. The larval zebrafish, a small model vertebrate, offers exceptional optical and genetic access to each population (Higashijima et al., 2000; Pittman et al., 2008; Schoppik et al., 2017). Further, the external development and anatomical simplicity of larval zebrafish (Bagnall and Schoppik, 2018) affords a unique opportunity to elucidate common motifs across an entire sensorimotor circuit.

The vestibular sensory periphery and their central/motor target populations are divergently organized. Functional imaging of the vestibular ganglion established a rostrocaudal axis for nose-down/nose-up segregation, matched to hair cell position along the utricular macula (Tanimoto et al., 2022). Electron micrographs suggest that vestibular ganglion maturation proceeds in the lateral-to-medial direction (Liu et al., 2022), conversely to hair cell development (medial-to-lateral) (Baeza-Loya et al., 2025 preprint). In contrast, their central and motor neuron partners are temporally and functionally aligned along the dorsoventral axis: early-born neurons are dorsally positioned and mediate nose-up/eyes-down behavior, while their late-born, ventral counterparts perform the converse (nose-down/eyes-up) (Goldblatt et al., 2023; Greaney et al., 2017). Notably, differentiating sensory progenitors are pushed dorsally as they delaminate from the ventral otic floor (Vemaraju et al., 2012), potentially decoupling functional topography from spatiotemporal development in the vestibular ganglion. Even still, dorsal vestibular ganglion neurons include both nose-up and nose-down subtypes (Liu et al., 2022; Tanimoto et al., 2022), underscoring the point that circuit assembly cannot be solved using a 'dorsal-to-dorsal' spatial map alone. Recent work uncovered that time organizes peripheral inputs to the vestibular ganglion (Liu et al., 2022) and, for other peripheral sensory afferents, central targeting (Pujol-Martí et al., 2012). We reasoned that, as suggested in Pujol-Martí et al. (2012) and Liu et al. (2022), time may also be the 'missing' variable that aligns the vestibular periphery with

downstream targets, and the complex task of polysynaptic partner matching achieved simply by assembling neurons with common temporal availability.

To determine if a common temporal motif could organize up/down channels across the entire vestibulo-ocular reflex circuit, we studied how the vestibular ganglion develops in space and time. Anatomical birthdating revealed that ganglion neurons emerged along dorsoventral and rostrocaudal gradients in time. Bipolar afferent projections to peripheral and central targets followed somatic development. With electron microscopy reconstructions, acute optical lesions and calcium imaging, we discovered that the earliest-born ganglion neurons form functional connections with temporally-matched central cohorts. Collectively, our findings support that the vestibular ganglion develops along a temporal axis shared with its downstream central and motor counterparts, where early-born neurons preferentially contribute to a specific channel (nose-up). Our findings complete the characterization of vestibulo-ocular reflex circuit development, revealing a unified spatiotemporal pattern to circuit assembly that spans peripheral sensory, central and motor areas.

## RESULTS

### Tg(-6.7Tru.Hcrtr2:GAL4-VP16) provides reliable access to the vestibular (statoacoustic) ganglion neurons that mediate vertical gaze stabilization

In zebrafish, the vestibular sensory neurons that relay utricular pitch-tilt (nose-up/nose-down) sensation are located in the anterior statoacoustic ganglion ('vestibular ganglion') (Sapède and Pujades, 2010; Taberner et al., 2017) (Fig. 1A). Past *in vivo* developmental studies used a transgenic line, *Tg(-17.6isl2b:GFP)*, that labels the vestibular ganglion with a fluorescent reporter driven by the *isl2b* promoter (Dyballa et al., 2017; Pittman et al., 2008; Roberts et al., 2017; Sapède et al., 2012; Vemaraju et al., 2012; Zecca et al., 2015). Functional topography was assessed using *Tg(myo6b:jGCaMP7f);Tg(myo6b:tdTomato)* (Tanimoto et al., 2022; Toro et al., 2015). However, neither lines permit tagging subsets of neurons across development.

We instead leveraged the *Tg(-6.7Tru.Hcrtr2:GAL4-VP16)* transgenic line (Lacoste et al., 2015; Schoppik et al., 2017), which labels a contiguous population whose location is consistent with the vestibular ganglion (Taberner et al., 2017; Sapède and Pujades, 2010) and that co-localizes with *Tg(-17.6isl2b:GFP)* (Fig. 1A). Electron microscopy (EM) reconstructions (Liu et al., 2022) and tracing from the utricular macula (Sapède and Pujades, 2010) support that vestibular ganglion soma are unlikely to be intermingled with other neuronal types. Thus, we interpret that the contiguous *Tg(-6.7Tru.Hcrtr2:GAL4-VP16)*-labeled population is specific to vestibular ganglion neurons.

To evaluate how efficiently and reliably *Tg(-6.7Tru.Hcrtr2:GAL4-VP16)* labels the vestibular ganglion, we used two existing references: the transgenic *Tg(-17.6isl2b:GFP)* (Fig. 1B) for *in vivo* comparison and an existing EM atlas of utricular-innervated vestibular ganglion neurons (Liu et al., 2022; Jia and Bagnall, 2022) (Fig. 1C). We mapped the spatial location of *Tg(-6.7Tru.Hcrtr2:GAL4-VP16)*, *Tg(-17.6isl2b:GFP)* and EM-labeled vestibular ganglion neurons to a common reference framework (Fig. 1D-G). Mapping was performed at 5 days post-fertilization (dpf), when vertical vestibulo-ocular reflex circuit anatomy is stable (Schoppik et al., 2017) and tilt-evoked eye rotation behavior is directionally appropriate (Bianco et al., 2012; Leary et al., 2025).

We first evaluated the density of labeled neurons across all three datasets (Fig. 1H). We observed a similar number of *Tg(-6.7Tru.Hcrtr2:GAL4-VP16)*-labeled neurons (median, 69±15 neurons/hemisphere) as in *Tg(isl2b:GFP)* (median, 73±12 neurons/

hemisphere; $P_{ranksum}$=0.55). However, both lines labeled fewer neurons than the EM reconstruction (106 utricular-innervated neurons; N=1 larva; $P_{Kruskal–Wallis}$=0.39). Other estimates indicate that the vestibular ganglion may contain as many as 100-130 neurons (Liu et al., 2022; Vemaraju et al., 2012). As neurogenesis continues through adulthood (Schwarzer et al., 2020), the discrepancies in neuronal density could arise from transgenic biases in labeling post-mitotic versus all (including immature) ganglion neurons.

We next evaluated the spatial distributions of labeled neurons (Fig. 1I). We compared our two *in vivo* datasets: *Tg(-6.7Tru.Hcrtr2: GAL4-VP16)* and *Tg(-17.6isl2b:GFP)*. Neurons in both lines spanned ~50 µm in the dorsoventral and rostrocaudal axes (dorsoventral: *hcrtr2*, 53±3 µm; *isl2b*, 51±3 µm; rostrocaudal: *hcrtr2*, 54±3 µm; *isl2b*, 53±2 µm) and 60 µm in the mediolateral axis. The distribution of labeled neurons was statistically similar across lines (dorsoventral: $P_{KS}$=0.86; rostrocaudal: $P_{KS}$=0.92; mediolateral: $P_{KS}$=0.99). Neurons mapped from EM reconstructions spanned a comparable, though slightly smaller, spatial extent: 47 µm dorsoventrally, 40 µm rostrocaudally and 40 µm mediolaterally. We interpret the minor size differences (~10 µm) between *in vivo* and EM-mapped neurons as fixation artifacts in the latter. Next, in both of our *in vivo* transgenic maps, we observed a correlation between medial and caudal position (Fig. 1G; r=−0.57). However, the EM reconstruction suggests that vestibular ganglion neurons are evenly distributed across the rostrocaudal axis, with a medial bias not evident in our transgenic maps (Fig. 1G,I). The medial bias suggests the presence of a rostromedial domain that is not labeled by either *Tg(-6.7Tru.Hcrtr2:GAL4-VP16)* or *Tg(-17.6isl2b:GFP)*, and which could include the additional ~30 neurons, whether post-mitotic or immature, that were not observed in our *in vivo* neuronal counts.

Lastly, given the tight relationships between vestibular ganglion geometry and functional organization (Liu et al., 2022; Tanimoto et al., 2022), we considered how vestibular ganglion expression in *Tg(-6.7Tru.Hcrtr2:GAL4-VP16)* compared to the *Tg(myo6b: jGCaMP7f);Tg(myo6b:tdTomato)* line that was previously described (Tanimoto et al., 2022). *Tg(-6.7Tru.Hcrtr2:GAL4-VP16)* and *Tg(isl2b:GFP)* appear to span a comparable spatial extent around the utricle as *Tg(myo6b:jGCaMP7f);Tg(myo6b:tdTomato)*. However, expression may be more inconsistent along the spatial edges (10-20 µm) and with approximately half the labeling density as *Tg(myo6b:jGcamp7f);Tg(myo6b:tdTomato)* (personal communication, M. Tanimoto). The comparable spatial extents suggests that *Tg(-6.7Tru.Hcrtr2:GAL4-VP16)* approximates, but might not comprehensively portray, spatially-linked developmental patterns.

Together, our reference mapping across three datasets highlights the importance and challenges associated with aligning variable anatomical geometry across transgenic lines and studies. We conclude that neither *Tg(-6.7Tru.Hcrtr2:GAL4-VP16)* nor *Tg(isl2b: GFP)* comprehensively accesses all utricular-innervated vestibular ganglion neurons; we speculate that the rostromedial domain, which may contain the latest-born neurons, may be unincorporated in these transgenics. Nevertheless, *Tg(-6.7Tru.Hcrtr2:GAL4-VP16)* provides comparable access to the vestibular ganglion as *Tg(isl2b:GFP)*, the most comprehensively-characterized *in vivo* transgenic line used to date, and to the majority of the vestibular ganglion. The implications of potentially inconsistent labeling of the vestibular ganglion are noted where appropriate below.

### Developmental time organizes a dorsoventral map in the vestibular ganglion

To tag vestibular ganglion neurons during development, we used *Tg(-6.7Tru.Hcrtr2:GAL4-VP16)* to express a photoconvertible

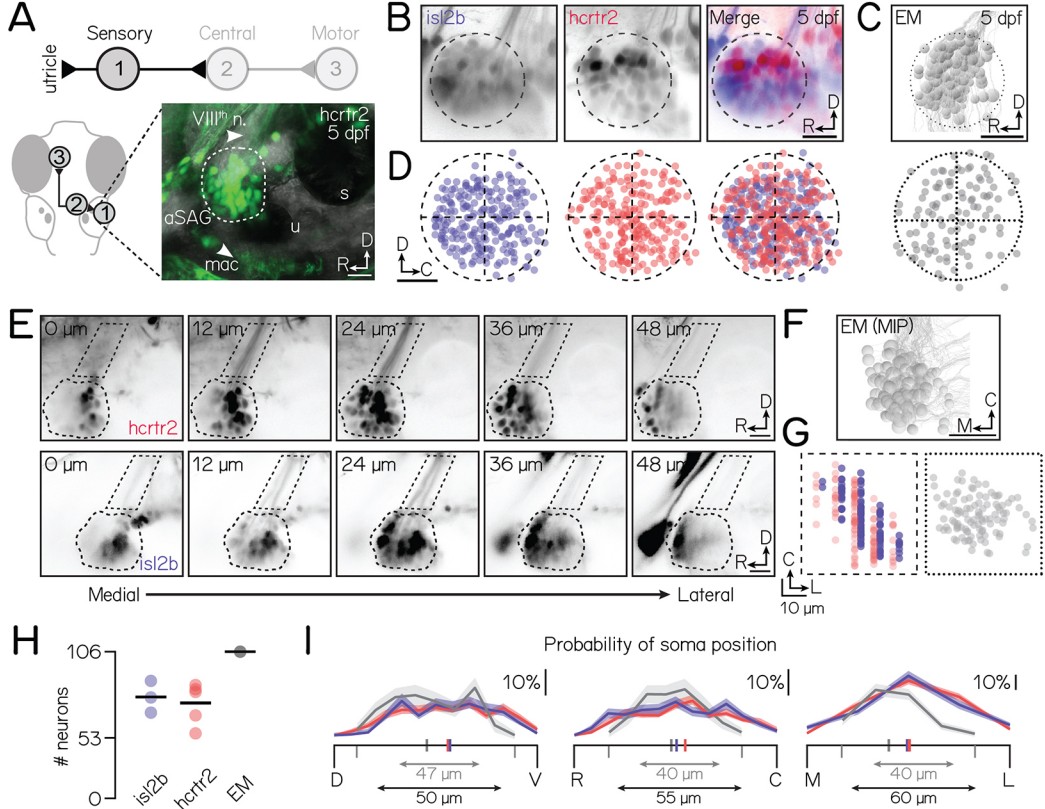

**Fig. 1. *Tg(-6.7Tru.Hcrtr2:GAL4-VP16)* and *Tg(-17.6isl2b:GFP)* provide comparable but partially incomplete genetic access to vestibular sensory neurons in the anterior statoacoustic ganglion.** (A) Schematic of information flow (top and left) in the vestibulo-ocular reflex circuit. Bipolar sensory neurons in the vestibular (anterior) ganglion are highlighted as the population of focus. Right inset shows transmitted light and fluorescent micrograph image of the vestibular ganglion, labeled in green by *Tg(-6.7Tru.Hcrtr2:GAL4-VP16);Tg(UAS-E1b:Kaede)*. White arrowheads point to bipolar afferents to the utricular macula (mac, bottom) and VIIIth nerve (top). Dashed line outlines the anterior (vestibular) ganglion. u, utricle; s, saccule; aSAG, anterior statoacoustic ganglion. (B) Visualization of vestibular ganglion neurons labeled by *Tg(-17.6isl2b:GFP)* (left; blue) and *Tg(-6.7Tru.Hcrtr2:GAL4-VP16)*; *Tg(UAS-E1b:Kaede)* (middle; red) in the same larva (sagittal view). Black dashed circle outlines the extent of the vestibular ganglion. (C) Electron microscopy (EM) reconstruction from Liu et al. (2022) of all utricular-innervated vestibular ganglion neurons in one larva at 5 dpf (sagittal view). Dotted line bounds the ganglion. (D) Soma position of mapped neurons within the boundaries of the vestibular ganglion, in larvae that expressed either *Tg(-17.6isl2b: GFP)* (n=220 neurons from N=3 larvae), *Tg(-6.7Tru.Hcrtr2:GAL4-VP16);Tg(UAS-E1b:Kaede)* (n=345 neurons from N=5 larvae; a subset of n=220 randomly selected neurons are shown) or EM reconstructed (n=106 neurons from N=1 larva). The vertical and horizontal dashed lines over blue/red show the rostrocaudal and dorsoventral midpoints based on the full spatial extent of vestibular ganglion neurons observed in the n=345 neurons from N=5 *Tg(-6.7Tru.Hcrtr2:VP16-GAL4)* larvae (Materials and Methods). Dashed lines and midpoints over gray are based on the EM-specific bounding box shown in C. (E) Representative images of the vestibular ganglion in *Tg(-6.7Tru.Hcrtr2:GAL4-VP16)* (top) and *Tg(-17.6isl2b:GFP)* (bottom) across five mediolateral subdivisions (orthogonal axis to D). Black dashed lines outline ganglion neuron somata (circle) and the VIIIth nerve (rectangle). (F) Maximum intensity projection of the EM-reconstructed ganglion neurons in an axial view, emphasizing the additional medial density. (G) Soma position of the neurons mapped in D, projected in the orthogonal (mediolateral) axis (left). Maps from genetically-labeled neurons (left) localize neurons to one of the five mediolateral subdivisions shown in E; right, EM reconstruction from F. (H) Number of neurons counted from each fish (dot). Solid bar shows median across larvae for isl2b and hcrtr2. (I) Probability of soma position in each spatial axis for all neurons mapped. Solid and shaded lines show the mean and standard deviation, respectively, using bootstrapped estimates of variance (Materials and Methods). Short vertical lines indicate the median position of neurons from each dataset. Gray/black vertical ticks delineate the spatial extent of the EM/transgenic volumes; black vertical lines are 10% probability. 'isl2b' refers to *Tg(-17.6isl2b:GFP)* and 'hcrtr2' refers to *Tg(-6.7Tru.Hcrtr2:GAL4-VP16)*. Scale bars: 20 µm.

reporter, Kaede (Scott et al., 2007), for permanent optical labeling of experimenter-defined temporal cohorts (Caron et al., 2008; Greaney et al., 2017; Goldblatt et al., 2023). We tagged neuronal cohorts born at intermediate stages 22-72 h post-fertilization (hpf) (Fig. 2A), the time course over which *isl2b*+ vestibular ganglion neurons differentiate in larval zebrafish (Haddon and Lewis, 1996; Vemaraju et al., 2012). We imaged vestibular ganglion neurons at 5 dpf. Tagged neurons (i.e. born before the time of Kaede photoconversion) were readily identifiable by the presence of red, converted Kaede; neurons born after conversion contained only green, unconverted protein (Fig. 2B). The number of tagged neurons progressively increased from 22 to 72 hpf, with a transient surge at ~33 hpf (N=5 converted larvae/time point; Fig. 2C;

Table 1). This transient phase could reflect the differentiation of neurons that are not incorporated in our genetic line, such as the rostromedial sector (Fig. 1G). Most neurons (91% of the number observed in our non-birthdated reference dataset, n=345 neurons/ N=5 fish shown in Fig. 1D,H) evaluated at 5 dpf were born by 72 hpf, in line with previous reports (Vemaraju et al., 2012).

To evaluate birthdate-linked spatial organization, we compared the location of the earliest- and latest-born neurons. By the chronological midpoint (48 hpf) of our observed differentiation window (22-72 hpf), the majority of labeled ganglion neurons were already born (~75%) (Table 1). Instead, the midpoint where 50% of neurons were born was 33-36 hpf (Table 1). We thus use 36 hpf as a reference midpoint to describe early- and late-born neurons. We

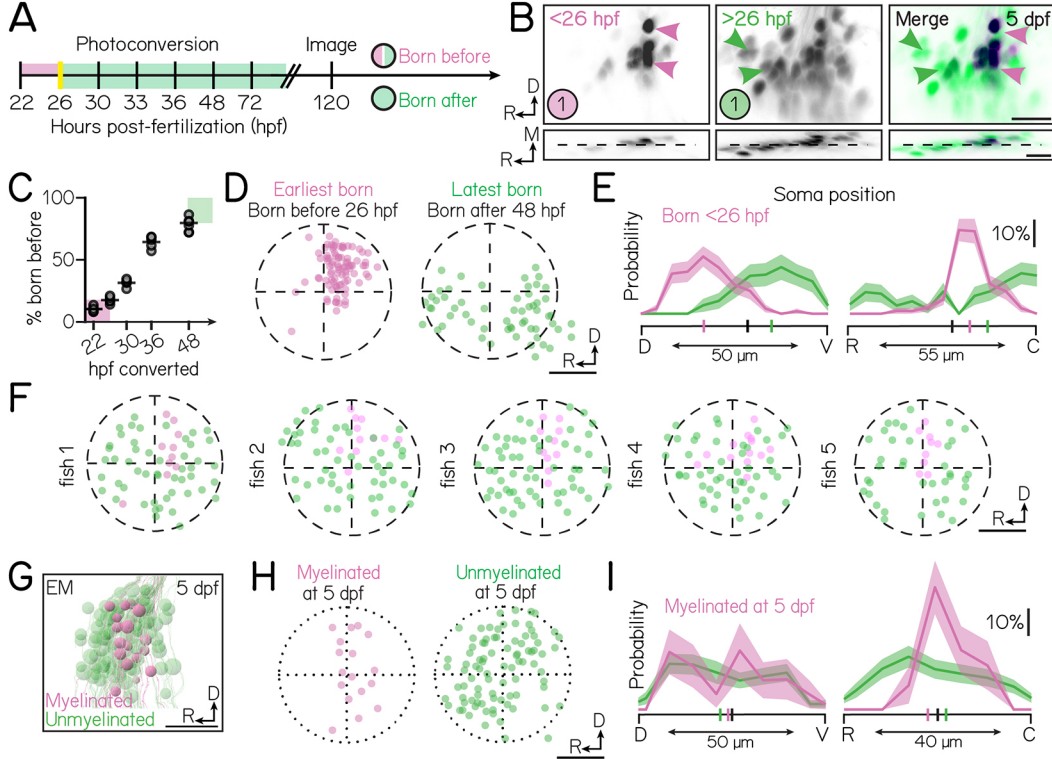

**Fig. 2. Early- and late-born cohorts of vestibular ganglion neurons are dorsoventrally opposed.** (A) Timeline of birthdating and imaging experiments. Time of differentiation is estimated by the presence of red and green Kaede (born before the time of photoconversion) or green Kaede only (born after photoconversion). Timeline shows all experimenter-defined conversion points; each larva is converted only once. Yellow line highlights 26 hpf as the example time point shown in B. (B) Example vestibular ganglion neurons (1) born before (left; magenta arrowheads) or after (middle; green arrowheads) 26 hpf (right: merge). Neurons are visualized by *Tg(-6.7Tru.Hcrtr2:GAL4-VP16);Tg(UAS-E1b:Kaede)* fish imaged at 5 dpf. The 'born before 26 hpf' image shown was thresholded to remove background fluorescence (Materials and Methods). Top: Sagittal maximum intensity projection used for mapping. Bottom: The same image in an orthogonal plane highlighting the mediolateral location of early-born neurons. (C) Percent of neurons in each fish that were born before each time point. Black lines indicate the mean number across individual fish (circles). *N*=5 fish for each age. Magenta and green shaded rectangles highlight the number of neurons born at the time points selected in D. (D) Location of neurons born before 26 hpf (left; 'earliest-born' cohort) or after 48 hpf (right; 'latest-born' cohort). Number of mapped neurons: early, *n*=88 neurons from *N*=10 fish (22 hpf and 26 hpf time points combined); late, *n*=58 neurons from *N*=5 fish. Dashed horizontal and vertical reference lines indicate the rostrocaudal and dorsoventral midpoints of the vestibular ganglion shown in Fig. 1D. (E) Probability of soma position for the neurons shown in D. Short vertical axis lines indicate the median position for control (black), early-born (magenta) or late-born (green) neurons. Solid and shaded lines shown the mean and standard deviation, respectively, using bootstrapped estimates of variance (Materials and Methods). (F) Location of neurons from *N*=5 randomly selected individual fish from the *N*=10 larvae aggregated in D. (G) EM reconstruction of myelinated (presumptive early-born neurons; magenta) and unmyelinated (green) neurons from one fish at 5 dpf. Data is reconstructed from Liu et al. (2022) contains the same neurons shown in Fig. 1C. (H) EM-reconstructed neurons, mapped to the same sagittal view framework as in D and F, to show the comparative localization of myelinated neurons at 5 dpf. (I) Probability of soma position for myelinated and unmyelinated neurons from the EM reconstruction. Solid and shaded lines shown the mean and standard deviation, respectively, using bootstrapped estimates of variance. Scale bars: 20 µm. Full statistics are shown in Table 2.

evaluated an early-born (before 26 hpf) and late-born cohort (after 48 hpf), collectively comprising half (41%) of all vestibular ganglion neurons (Table 1). Strikingly, these cohorts were differentially organized along the dorsoventral axis (Fig. 2D). We quantified the probability of observing early- and late-born soma in each spatial axis (Fig. 2E). Early and late cohort position was significantly different along the dorsoventral ($P_{KS}=1.1\times10^{-16}$) and rostrocaudal ($P_{KS}=1.7\times10^{-4}$) axes. Qualitatively, the earliest-born cells may be medially positioned (Fig. 1B). However, statistically significant mediolateral organization was not observed ($P_{KS}=0.25$). As some medial neurons may not be labeled in *Tg(-6.7Tru.Hcrtr2: GAL4-VP16)* (Fig. 1G) and the mediolateral axis was orthogonal to our imaging and mapping plane, we cannot rule out the possibility of additional mediolateral organization.

Importantly, the early-born cohort appeared to be caudal of center relative to our non-birthdated reference dataset (Fig. 1D,I) ($P_{KS}=1.4\times10^{-11}$) in aggregate maps (*N*=10 larvae: 5 tagged at 22 hpf and 5 tagged at 26 hpf; Fig. 2D,E). As the vestibular

ganglion is rostrocaudally organized (rostral, nose-down; caudal, nose-up) (Liu et al., 2022; Tanimoto et al., 2022), we more closely examined this caudal bias. Individual maps from *N*=5 larvae (Fig. 2F) suggested that most neurons born before 26 hpf lay on or very near to the rostrocaudal midpoint of our (non-birthdated) reference dataset. As a secondary assessment, we re-evaluated the EM reconstruction, which had inferred neuronal age using myelination at 5 dpf (Fig. 2G). We mapped myelinated (magenta; presumably earliest-born) and un-myelinated (green; late-born) neurons to our reference framework (Fig. 2G). As in our data, myelinated neurons were distributed largely around the rostrocaudal midpoint, with a statistically significant caudal bias (myelinated versus unmyelinated; Fig. 2H,I; rostrocaudal: $P_{KS}=0.01$, dorsoventral: $P_{KS}=0.93$). Though myelinated neurons were equally observed across the dorsal and ventral domains, this is not incongruent with our *in vivo* birthdating findings; myelination status at 5 dpf likely includes neurons born well-after 26 hpf. Together, *in vivo* and EM maps suggest that the earliest-born

**Table 1. Number of statoacoustic ganglion neurons converted across developmental time points**

| Age converted | Number of cells/fish born before | Approximate % of control born before | Number of cells/fish born after |
|---|---|---|---|
| 22 | 7±2 | 10% | 59±7 |
| 26 | 11±2 | 16% | 49±7 |
| 30 | 21±2 | 30% | 44±5 |
| 33 | 42±2 | 61% | 36±5 |
| 36 | 40±5 | 58% | 22±4 |
| 48 | 52±8 | 75% | 12±2 |
| 60 | n/a | – | 3±1 |
| 72 | 63±5 | 91% | 0.5±0.5 |
| Non-birthdated reference dataset | 69±15 | | |

Data shown as mean±s.d. number of neurons born before or after each time point from N=5 hemispheres/time point. 'Non-birthdated reference dataset' refers to the median number of neurons per fish observed from a reference dataset of the n=345 neurons vestibular ganglion neurons mapped from N=5 larvae from *Tg(-6.7Tru.Hcrtr2:VP16-GAL4);Tg(UAS-E1b:Kaede)* (Fig. 1D,G).

ganglion neurons may in fact be caudal biased, supporting a nose-up designation.

The spatial patterns we observed (dorsal, caudal) could reflect discrete spatiotemporal windows of neurogenesis or arise from gradients generated continuously across time. Evidence that dorsal and/or caudal neurons continue to accumulate at later developmental stages would support the latter. To differentiate these hypotheses, we evaluated the location of neurons from our *in vivo* dataset born before (Fig. 3A) and after (Fig. 3B) all seven of our converted time points (N=5 larvae/cohort; full statistics in Table 2). The dorsal bias we observed persisted until ~36 hpf

(36 hpf versus 33 hpf: $P_{\text{ANOVA multiple comparisons test}}$=0.20; Fig. 3C). Qualitatively, by 48 hpf, nearly all 'born after' neurons were located ventral to the midpoint of the ganglion (Fig. 3B). We interpret that dorsal and ventral neurons are largely born in opposing temporal windows, offset by ~8-12 h: dorsal neurons from 22-48 hpf and ventral neurons from 30-72 hpf.

As the earliest born 30% of the vestibular ganglion may lie caudal of center (Fig. 2E), we next asked if progressive rostrocaudal development also anticipates functional topography. We evaluated the rostrocaudal distribution of neurons at each stage relative to the spatial midpoint. The exclusive caudal bias to early-born neurons was discontinued by 30 hpf (Fig. 3A), when rostral neurons first became apparent. After 33 hpf (remaining 70% of neurons), rostral and caudal neurons emerged contemporaneously (33 hpf versus 30 hpf: $P_{\text{ANOVA multiple comparisons test}}$=0.94; Fig. 3D; Table 2). Both caudal and rostral neurons remained unconverted after 60 hpf (Fig. 3B). While our dataset contains a low number of unconverted neurons at 72 hpf, the presence of rostral neurons at this age suggests that – unlike their dorsal counterparts – ventrorostral and ventrocaudal accumulate continuously from 33 hpf through late development.

We conclude that ganglion development progresses along a major dorsoventral and secondary rostrocaudal axis. The dorsocaudal location of the earliest-born cells may anticipate their functional (nose-up) topography. While our genetic line may not comprehensively access the rostromedial sector, that both ventrorostral and ventrocaudal neurons emerge contemporaneously through late development (72 hpf) suggests that spatial and functional organization are uncoupled after 33 hpf and for the vast majority (70%) of labeled neurons. Contrastingly, the clearly delineated

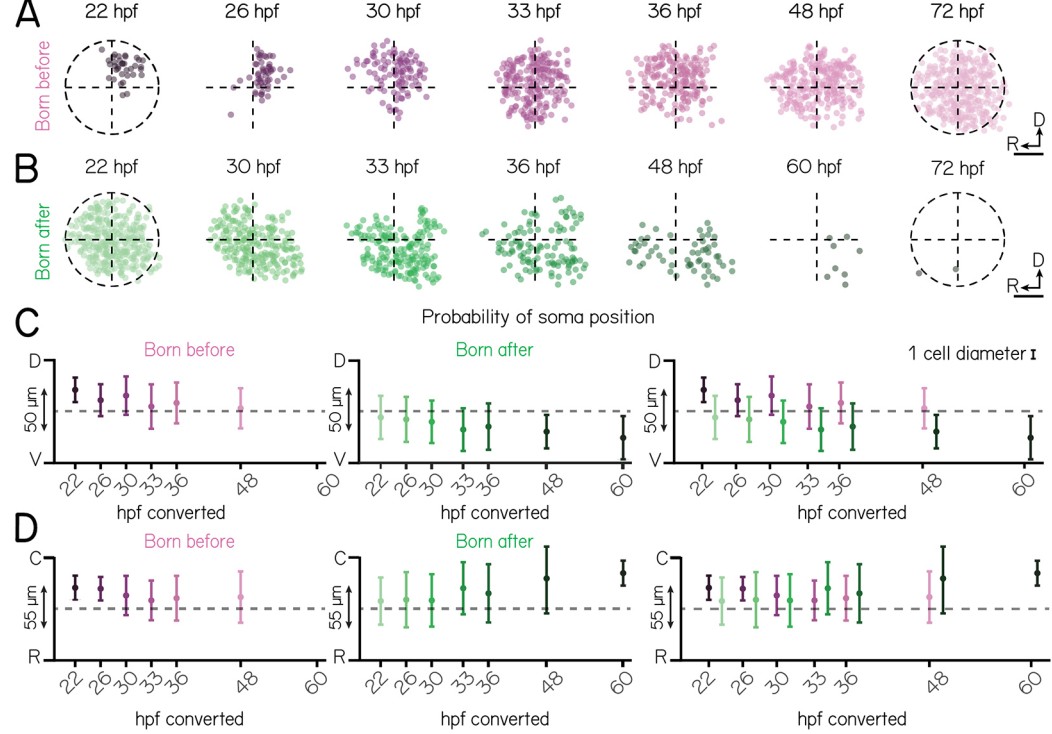

**Fig. 3. Position along the dorsoventral axis predicts late-born somatic accumulation within the vestibular ganglion.** (A) Soma position of neurons born before each time point (N=5 fish/time point). Dashed horizontal and vertical reference lines indicate the rostrocaudal and dorsoventral midpoints shown in Fig. 1D. (B) Soma position of neurons born after each time point (N=5 fish/time point). (C) Dorsoventral position of neurons born before (magenta) or after (green) each time point, shown as median/standard deviation across aggregated data. Horizontal dashed line shows the midpoint. (D) Rostrocaudal position of neurons born before/after each time point, shown as the median/standard deviation across aggregated data. Full statistics are shown in Table 2.

**Table 2. Statistical comparisons of temporal cohort somatic organization**

| Age converted | Median position born before (µm) | Median position born after (µm) | $p_{KS}$ born before versus born after | $P_{ANOVA,multcompare}$ born before versus previous age | $P_{ANOVA,multcompare}$ born after versus previous age |
|---|---|---|---|---|---|
| Dorsoventral | | | | | |
| 22 | 17.3±7.2 | 33.5±12.7 | 9.9E-10 | – | – |
| 26 | 23.4±9.4 | 34.7±13.3 | 7.6E-07 | 0.50 | 0.99 |
| 30 | 20.7±11.3 | 36.1±12.5 | 2.3E-12 | 0.99 | 0.80 |
| 33 | 27.1±13.1 | 40.7±12.5 | 9.0E-09 | 5.2E-05 | 0.08 |
| 36 | 25.0±12.0 | 38.9±13.6 | 2.2E-11 | 0.20 | 0.97 |
| 48 | 28.2±11.8 | 41.9±9.9 | 7.2E-09 | 0.18 | 0.34 |
| 60 | n/a | 45.5±12.7 | n/a | – | 0.99 |
| 72 | 34.5±13.9 | 53.3±2.0 | 0.04 | – | 0.99 |
| Rostrocaudal | | | | | |
| 22 | 42.7±7.1 | 34.9±13.9 | 7.6E-06 | – | – |
| 26 | 42.2±6.9 | 35.7±16.2 | 2.2E-08 | 0.99 | 0.99 |
| 30 | 38.2±11.5 | 35.3±15.4 | 0.08 | 0.10 | 0.99 |
| 33 | 35.3±11.7 | 42.4±15.3 | 6.6E-07 | 0.94 | 0.48 |
| 36 | 36.6±13.2 | 39.4±17.1 | 1.1E-04 | 0.99 | 0.99 |
| 48 | 37.2±15.1 | 48.2±20.6 | 7.9E-04 | 0.97 | 0.99 |
| 60 | n/a | 51.3±7.3 | n/a | – | 0.66 |
| 72 | 38.2±14.1 | 21.8±17.1 | 0.28 | – | 0.24 |
| Mediolateral | | | | | |
| 22 | 36.0±7.7 | 36.0±10.5 | 0.01 | – | – |
| 26 | 36.0±9.1 | 36.0±11.6 | 0.03 | 0.99 | 0.99 |
| 30 | 36.0±10.6 | 36.0±12.3 | 0.99 | 0.04 | 0.99 |
| 33 | 36.0±10.7 | 36.0±10.8 | 0.005 | 0.99 | 0.48 |
| 36 | 36.0±10.7 | 36.0±10.9 | 0.06 | 0.99 | 0.99 |
| 48 | 36.0±9.7 | 36.0±10.5 | 0.52 | 0.97 | 0.99 |
| 60 | – | 36.0±5.6 | – | – | 0.66 |
| 72 | 36.0±10.1 | 30±8.5 | 0.73 | – | 0.24 |

Median±s.d. positions are shown as microns from the dorsal- or rostral-most edges (0 µm) of the vestibular ganglion. Data are shown as an aggregate of all neurons mapped across N=5 hemispheres/time point.

dorsoventral patterns we observe elevates the dorsoventral axis as a temporally-enduring predictor of vestibular ganglion accumulation.

## Bipolar afferent organization follows spatial patterns of somatic development

Somatic topography anticipates axonal connectivity (Dasen, 2009; Sürmeli et al., 2011; Balaskas et al., 2020; Goldblatt et al., 2023). Vestibular ganglion neurons send bipolar afferents ventrally to peripheral hair cells in the utricular macula and dorsally to central hindbrain targets via the VIIIth nerve (Sapède and Pujades, 2010; Liu et al., 2022) (Fig. 4A,B). We hypothesized that the orderly positioning of early-born vestibular ganglion neurons could spatially constrain the path of their projections within each bipolar afferent bundle. We therefore analyzed the bipolar afferents of the converted neurons described previously (Fig. 4C) along its width (rostrocaudal axis). Analysis in the dorsoventral axis was not possible, as projections inherently grow towards either dorsal or ventral targets. We evaluated the spatial distribution of converted axons (i.e. from neurons born before the time of conversion) at 5 dpf, reasoning that organization at 5 dpf reflects how axons become assembled into the VIIIth nerve.

We first evaluated peripheral projections to the utricular macula (Fig. 4D,E; N=3 hemispheres/time point). Converted axons from soma born before 22 hpf were only observed in the caudal macula ('born before' versus all fluorescence, rostrocaudal axis: $P_{KS}=5.3\times10^{-11}$). Conversion spread rostrally by 30 hpf and spanned the full rostrocaudal extent of the macula by 36 hpf ($P_{KS}=0.50$), matching the temporal emergence of rostrally-positioned somata. Next, we evaluated projections to the VIIIth nerve (Fig. 4F-H; N=3 hemispheres/time point). In the caudal half of the axon bundle, we observed a faint but visible converted axon

from soma born before 22 hpf. We observed stronger conversion in the caudal and center of the axon bundle from somata born before 26 hpf ($P_{KS}=0.016$). As with the macula, rostral spread across the VIIIth nerve bundle was evident by 30 hpf ($P_{KS}=0.69$). Together, we conclude that rostrocaudal organization within the utricular macula and VIIIth nerve follows neuronal birthdate: early-born soma (caudal) project caudally or center-caudally within each axon bundle.

## Early-born vestibular ganglion neurons assemble with spatiotemporally- and functionally-matched central neuron cohorts

Does birthdate anticipate synaptic assembly between partner populations across the developing vestibulo-ocular reflex circuit? Vestibular ganglion neurons relay tilt sensation to central (hindbrain) projection neurons in the tangential nucleus (Liu et al., 2022; Goldblatt et al., 2023), which are necessary and sufficient for vertical (nose-up/nose-down) gaze stabilization behavior (Bianco et al., 2012; Schoppik et al., 2017). Previously, we found that both birthdate and function segregate central vestibular neurons: early-born dorsal neurons are sensitive to nose-up tilts, and late-born ventral neurons are sensitive to nose-down tilts (Goldblatt et al., 2023). This pattern mirrors the dorsoventral development we observed for their vestibular ganglion inputs and occurs over a comparable temporal window (22-48 hpf) (Goldblatt et al., 2023). Thus, we predicted that the earliest-born vestibular ganglion neurons (dorsal, caudal, nose-up) would assemble with temporally- (early-born), spatially- (dorsal) and functionally- (nose-up) matched central targets.

We first evaluated whether vestibular ganglion afferents from early-born neurons were anatomically poised to wire to temporally-matched central targets. At 5 dpf, VIIIth nerve input from neurons

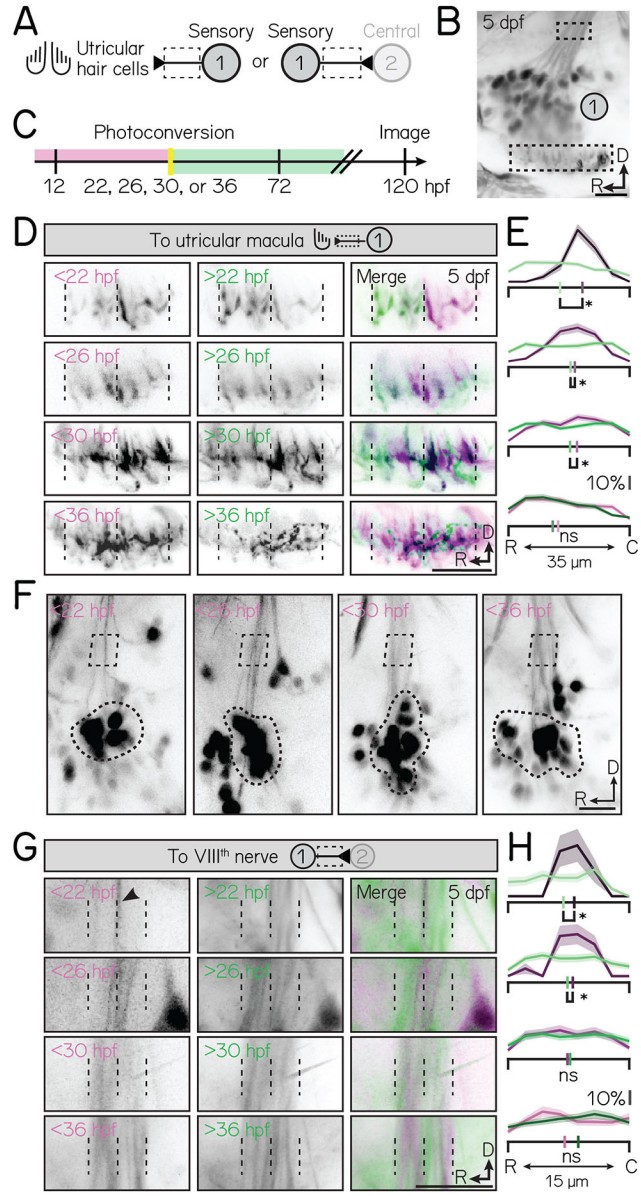

**Fig. 4. The development of bipolar afferents to local utricular targets and the VIIIth nerve mirrors somatic topography.** (A) Circuit schematic. Black dashed lines outline the circuit elements of focus. (B) Representative image of the vestibular ganglion visualized using *Tg(-6.7Tru.Hcrtr2:GAL4-VP16)*;*Tg(UAS-E1b:Kaede)*. Rectangles outline bipolar afferents to the utricular macula (bottom) and VIIIth nerve (top). (C) Timeline of birthdating experiments. Larvae were converted at only one time point (22, 26, 30 or 36 hpf) and imaged at 5 dpf. (D) Vestibular ganglion projections terminating at the utricular macula at four converted time points (left, converted Kaede; middle, unconverted Kaede; right, merge). As neurons continue to generate new, unconverted Kaede after the time of conversion, green projections may originate from both 'born before' and 'born after' neurons. Black dashed vertical line shows the approximate rostrocaudal extent and midpoint of vestibular ganglion projections based on total unconverted fluorescence (all projections). Data from one example fish/time point shown. (E) Probability of observing converted Kaede (magenta) along the rostrocaudal axis. Each distribution corresponds with the time point shown to the left (D). Green shows the probability of observing unconverted Kaede. Solid and shaded lines shown the mean and standard deviation, respectively. *N*=3 larvae quantified per time point. (F) Projections to the VIIIth nerve from converted vestibular ganglion somata. Dashed boxes outline the area of the projection bundle quantified in G. Dashed circular outline shows somata that were born before the converted time point and were above thresholded background levels (Materials and Methods). Non-converted, below-threshold soma fluorescence is an artifact of adjusting image brightness to visualize the fainter projection bundle. (G) Magnification of VIIIth nerve projections of one fish at one time point. Black arrowhead highlights a faint but visibly converted projection converted at 22 hpf. Black dashed lines show the rostrocaudal extent and midpoint. (H) Probability of observing converted (magenta) or unconverted (green) Kaede fluorescence along the rostrocaudal axis for each time point shown in G. Solid and shaded lines shown the mean and standard deviation, respectively. Scale bars: 20 µm. *P<0.05; n.s., not significant.

with the tangential nucleus, reasoning that some afferents may pass through the tangential nucleus en route to other synaptic targets. Still, tangential nucleus neurons that were synaptically coupled with myelinated ganglion afferents were preferentially dorsal and medial, aligning with its functional topography (Fig. 5D) (Goldblatt et al., 2023). Together, these anatomical observations support that early-arriving sensory input from the vestibular ganglion may preferentially target the earliest-born, nose-up tangential nucleus neurons.

To test our anatomical predictions, we evaluated whether early-born neurons in the vestibular ganglion and tangential nucleus are functionally connected. We leveraged an existing acute optical lesion (Goldblatt et al., 2023; Hamling et al., 2024; Schoppik et al., 2017) and calcium imaging (Goldblatt et al., 2023; Hamling et al., 2023) paradigm (Fig. 5E,F). First, we birthdated larvae at 26 hpf to tag the earliest-born 15% of vestibular ganglion soma and visualized neurons at 5 dpf. Lesioning early-born soma was not technically feasible due to their close proximity to the dorsal VIIIth nerve bundle and local vasculature. Instead, we unilaterally lesioned somata of the 85% of ganglion neurons born after 26 hpf (Fig. 5E), leaving only the early-born (dorsocaudal, nose-up) neurons available to relay tilt sensation. Somatic lesions secondarily degrade the axon projections of targeted neurons (Goldblatt et al., 2023; Hamling et al., 2024; Schoppik et al., 2017). Thus, by targeting our lesions to neuronal somata, our approach is specific to the late-born neurons and their projections. To evaluate changes in central responses to tilt sensation after ablating late-born ganglion neurons, we used Tilt-In-Place Microscopy (TIPM) (Hamling et al., 2023). Briefly, TIPM reports changes in a genetically-encoded calcium indicator, GCaMP6s (Chen et al., 2013), in central neurons after delivering nose-up and nose-down pitch tilts (Fig. 5F). At 5 dpf, the vestibular ganglion is the primary source of pitch tilt information (Goldblatt et al., 2023). Therefore, loss of central

born before 22 hpf exclusively targeted the early-born (dorsal) tangential nucleus (Fig. 5A). Thus, early-born (dorsal) vestibular ganglion neurons may innervate commonly positioned (early-born, dorsal) central neuron partners. However, *Tg(-6.7Tru.Hcrtr2: GAL4-VP16)* labels other cranial sensory ganglia that develop comparably early and project along the VIIIth nerve (Zecca et al., 2015).

To isolate afferents from the vestibular ganglion, we re-evaluated the EM dataset from Liu et al. (2022) and Jia and Bagnall (2022) (Fig. 1C), which had also reconstructed projections to central vestibular targets in the hindbrain. We again used myelination at 5 dpf to infer birth order. At 5 dpf, myelinated neurons preferentially localize to the lateral ganglion (Liu et al., 2022). Correspondingly, their afferents formed a lateral bundle at the proximal edge of the VIIIth nerve (Fig. 5B). However, at the hindbrain entry site, the afferent bundle inverted such that myelinated afferents innervated the medial (dorsal) tangential nucleus (Fig. 5B,C). We next restricted our analysis to vestibular ganglion afferents that were synaptically coupled

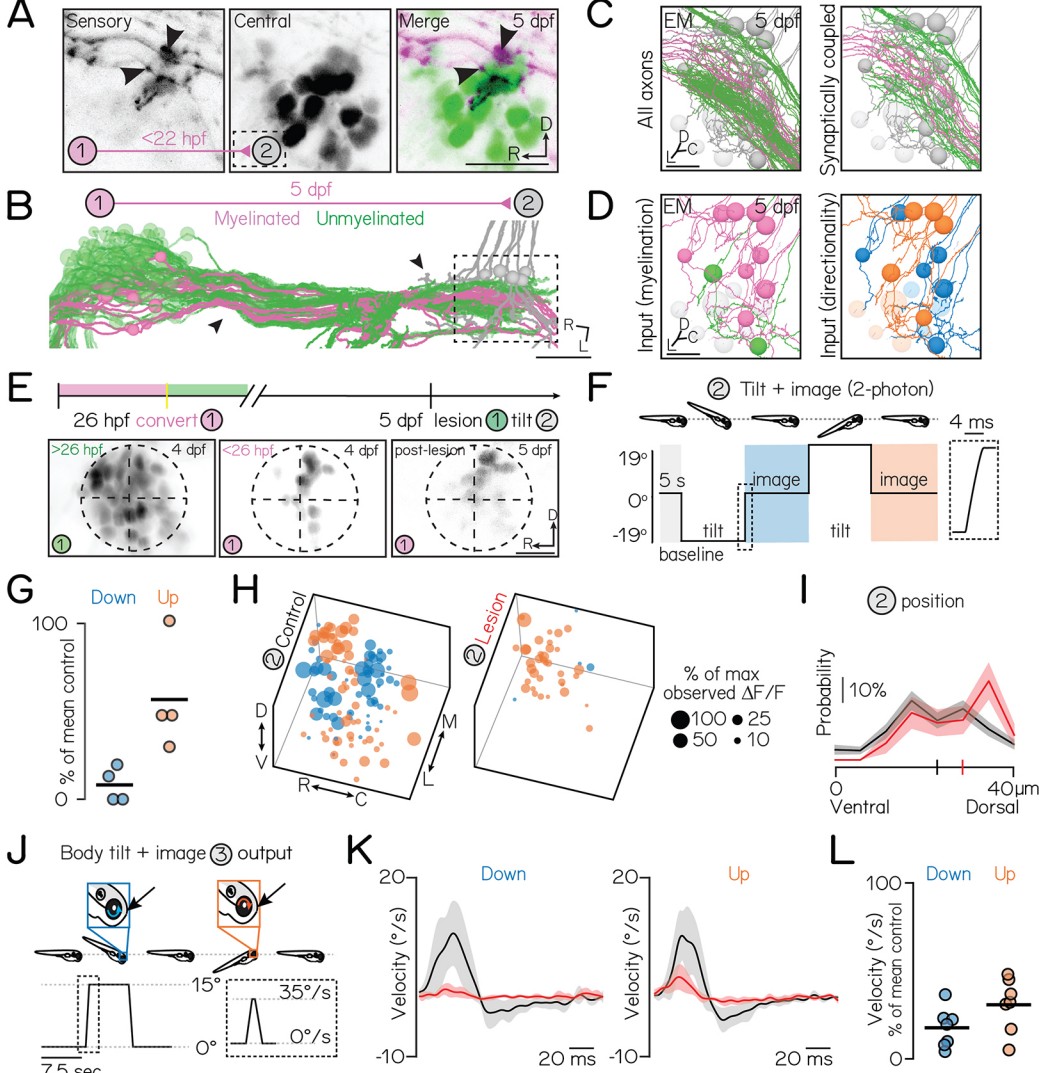

**Fig. 5. Early-born vestibular ganglion neurons assemble with spatiotemporally-matched central partners in the hindbrain.** (A) VIIIth nerve axons and tangential nucleus somata born before (left; magenta, 1) and after (middle; green, 2) 22 hpf. Black arrowheads point to VIIIth nerve innervation. Axons and neurons are visualized using *Tg(-6.7Tru.Hcrtr2:GAL4-VP16);Tg(UAS-E1b:Kaede)*. (B) EM reconstruction from Liu et al. (2022) of VIIIth nerve projections to the tangential nucleus at 5 dpf. Birth order is inferred from myelination (magenta, myelinated; green, unmyelinated). Black dashed box shows afferent organization at the tangential nucleus. Afferents from all *n*=106 reconstructed ganglion neurons are shown. (C) Pseudo-sagittal (orthogonal to B) projection of VIIIth nerve projections highlighting the dorsomedial entry point of myelinated ganglion afferents. Left image shows all ganglion afferents; right shows only afferents with synaptic connections onto the tangential nucleus. (D) EM reconstruction of the tangential nucleus (*n*=17 neurons) in the same view as C. Left: neurons are pseudocolored by whether they receive synaptic input from myelinated (magenta) or unmyelinated (green) vestibular ganglion afferents. Right: neurons are pseudocolored according to inferred directionality [nose-up, orange; nose-down, blue, (Liu et al., 2022)] of their vestibular ganglion inputs. Semi-transparent neurons (ventral) receive input from semicircular canal afferents but not utricular afferents, and may be included in the functional maps shown in H. (E) Top: Timeline of birthdating, lesion, and tilt experiments. Larve are converted at 26 hpf. At 5 dpf, unconverted vestibular ganglion neurons (green 1) are lesioned. Central projection neurons expressing GCaMP6s are imaged following tilt stimulation. Bottom: Vestibular ganglion neurons born after (left) or before (middle/right) 26 hpf, pre- (middle) and post- (right) lesion in a thresholded image (Materials and Methods). Circle and dashed lines from Fig. 1D. (F) Schematic of stimulation and calcium imaging paradigm. Shaded bars show imaging periods for baseline (gray), return from nose-down tilt (blue), and return from nose-up tilt (orange). Periods between shaded bars show time when the larvae is tilted away from the imaging plane. Inset shows time course of step to horizontal. (G) The number of tilt-responsive nose-up and nose-down neurons in a given larva (circles) after lesion of vestibular ganglion neurons born after 26 hpf, quantified as a percentage of the mean number of up/down neurons measured across *N*=4 control larvae. (H) Location of tangential nucleus neurons in control and lesioned larvae. Each circle is an individual cell, scaled by the magnitude of tilt-evoked fluorescence change. *n*=172 neurons from *N*=4 control hemispheres and *n*=115 neurons from *N*=4 lesioned hemispheres. (I) Distribution of central neuron somata in the dorsoventral axis in control (gray) and lesioned (red) larvae shown in H. (J) Top: vestibulo-ocular reflex trial schematic. Bottom: body tilt kinematics. (K) Average eye velocity following nose-up and nose-down body tilts in control (gray, *N*=6) and lesioned (red, *N*=7) larvae. Solid and shaded lines shown the mean and standard deviation, respectively. (L) Peak mean eye velocity after late-born ganglion lesion, shown as a percentage of the mean observed in control larvae. Scale bars: 20 μm.

neuron responses to tilt sensation should directly reflect loss of corresponding ganglion input.

We predicted that loss of late-born (ventral) vestibular ganglion inputs would eliminate pitch tilt responses in spatiotemporally-matched (late-born, ventral, nose-down) central neurons. We first imaged the tilt responses of *n*=172 tangential nucleus neurons from *N*=4 control hemispheres and *n*=115 neurons from *N*=4 hemispheres with acute, unilateral ganglion lesion. We observed a

substantial reduction in the number of tilt-responsive neurons (median control: 89.4±6.0% responsive; lesion: 36.9±21.4% responsive; $P_{ranksum}$=0.03), driven by a near total loss to nose-down tilts (nose-down: median control, $n$=8±5 neurons/fish; lesion: $n$=1±2 neurons/fish; $P_{ranksum}$=0.03; nose-up: control, $n$=17 ±2 neurons/fish; lesion: $n$=16±4 neurons/fish; $P_{ranksum}$=0.09; Fig. 5G). Central neurons that remained able to respond to tilt sensation were dorsally-shifted (median position: control, 25±8 μm; lesion, 30±7 μm, $P_{KS}$=0.01; Fig. 5H,I), consistent with the loss of ventral (nose-down) neurons (Goldblatt et al., 2023).

The loss of tilt sensitivity in ventral, nose-down central neurons should consequently impair upwards eye rotation behavior. To test this prediction, we leveraged an existing assay of vestibulo-ocular reflex behavior, which measures torsional eye rotation responses (eyes-up/eyes-down) following body tilts (nose-down/nose-up) (Fig. 5J) (Schoppik et al., 2017; Leary et al., 2025). We compared torsional eye rotation behavior after performing acute, bilateral lesions of either late-born ganglion neurons ($N$=6 fish) as before or the anterior lateral line ($N$=7 fish), which does not mediate vestibulo-ocular reflex behavior (control). As expected, late-born ganglion lesion significantly impaired eye rotation responses relative to control (Fig. 5K,L). Eye rotation responses were nearly entirely abolished to nose-down tilts ($P_{ranksum}$=0.002; peak angular velocity: 17.6% of mean control). We observed an additional loss of nose-up tilt responses ($P_{ranksum}$=0.001; peak angular velocity: 30.6% of mean control). These deficits are in line with the loss of functional sensitivity in the tangential nucleus.

Together, we interpret the absence of tilt responses in the ventral tangential nucleus and ensuing eye rotation deficits to be a direct consequence of loss of late-born (ventral) vestibular ganglion inputs. Correspondingly, the preservation of dorsal (nose-up) central neuron responses suggests a tight matchup in time between early-born (dorsocaudal, nose-up) ganglion neurons and their early-born (dorsal, nose-up) targets. Collectively, these experiments demonstrate that early-born ganglion neurons assemble with spatiotemporally-matched central neuron targets,

and that birth order is a continuous axis that links functional subtypes of vestibulo-ocular reflex neurons.

## DISCUSSION

To determine whether temporal organization could assemble a polysynaptic sensorimotor circuit, we studied how time comes to assemble the vestibular ganglion and its connectivity. We discovered that development proceeds along a dorsal-to-ventral and, secondarily, a caudal-to-rostral axis for a subset of early-born neurons, reminiscent of the ganglion's functional organization. Afferents from the vestibular ganglion systematically innervate their utricular and central (vestibular hindbrain) targets in a similar spatial and temporal progression. Intriguingly, these organizational motifs are shared across the entire vestibulo-ocular reflex circuit (Fig. 6). Indeed, our loss-of-function experiments establish that early-born vestibular ganglion neurons preferentially assemble with early-born central partners, just as early-born central neurons assemble with early-born motor neuron targets (Goldblatt et al., 2023). Taken together, our work reveals that the vertical vestibulo-ocular reflex circuit assembles sequentially in time, aligning sensory, central and motor output neurons into separate up and down channels. Our findings speak to the earliest steps in construction of an ancient, conserved and canonical sensorimotor reflex circuit.

### Limitations in genetic reagents and their implications for understanding the organization and specification of vestibular ganglion subtypes

Our characterization of vestibular ganglion anatomy using two transgenic lines and an existing electron microscopy dataset reveal the challenges associated with aligning organizational patterns across studies, particularly given the small size (50-60 μm) of the vestibular ganglion. Both transgenic reagents used in our study, $Tg(-6.7Tru.Hcrtr2:GAL4-VP16)$ and $Tg(isl2b:GFP)$ likely under-sample the vestibular ganglion in neuronal number (55-70% of EM-reconstructed neurons) and spatial distribution (rostromedial access). Across transgenics, additional bias could be introduced by the varying temporal integration of GAL4-UAS reagents compared

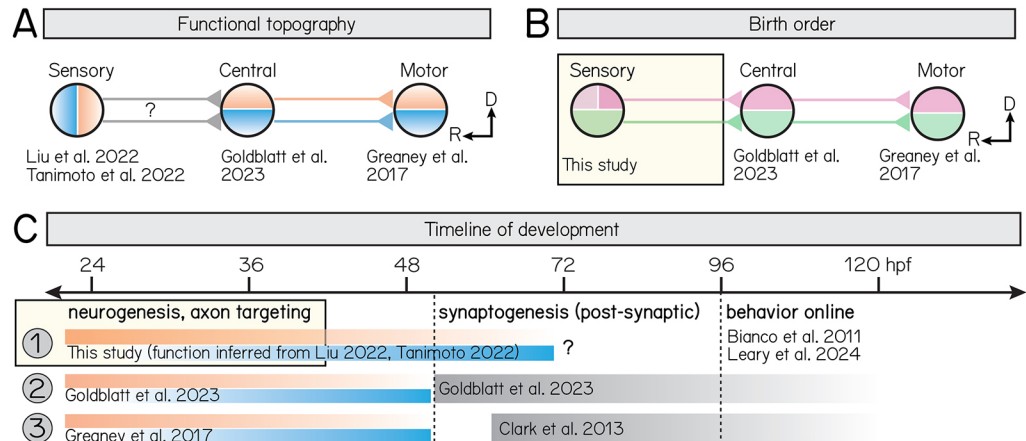

**Fig. 6. Birth order unifies vestibulo-ocular reflex circuit organization.** (A) Functional topography for each vestibulo-ocular reflex circuit population, adapted from previous papers (Liu et al., 2022, Tanimoto et al., 2022, Goldblatt et al., 2023, Greaney et al., 2017). Orange and blue represent nose-up and nose-down domains for each nucleus (circles) and their axonal projections (horizontal lines). (B) Birth order-related organization of each circuit population, adapted from the present study (yellow box) and Goldblatt et al. (2023) and Greaney et al. (2017). Magenta and green represent early- and late-born cohorts, respectively. (C) Timeline of key developmental events for each circuit population in zebrafish. Data are from this study (yellow box) and from previous papers (Goldblatt et al., 2023, Greaney et al., 2017, Clark et al., 2013, Bianco et al., 2012, Leary et al., 2025). Up/down function in the vestibular ganglion is inferred based on correlations between anatomical position (this study) and functional topography (Liu et al., 2022, Tanimoto et al., 2022). The time course of sensory-central synaptogenesis is yet to be determined in zebrafish.

to direct promoter-reporter fusion lines. Indeed, a different fusion line, *Tg(myo6b:jGCaMP7f);Tg(myo6b:tdTomato)* (Toro et al., 2015; Tanimoto et al., 2022), appears to access a similarly extended population of ganglion neurons. Further, vestibular ganglion progenitors continuously proliferate through adulthood (Schwarzer et al., 2020), generating differentiated neurons well-past our study endpoint (5 dpf). Intriguingly, the vestibular ganglion becomes myelinated in the lateral-to-medial direction (Liu et al., 2022), suggesting that the youngest (unmyelinated) neurons could occupy the rostromedial sector that was not labeled by either *Tg(-6.7Tru.Hcrtr2:GAL4-VP16)* or *Tg(isl2b:GFP)*. Alternatively, more recent work discovered that hair cell inputs develop in the converse direction (medial-to-lateral) (Baeza-Loya et al., 2025 preprint), in line with our observed medial localization to early-born ganglion neurons. Future studies could resolve these divergent observations by tracking the migration patterns of the same vestibular ganglion neurons across development and life (Schwarzer et al., 2020), as recent work tracked progenitor migration linked to coarse (anteroposterior) ganglion domain formation (Dyballa et al., 2017).

The caveats in our genetic access to the vestibular ganglion merits caution in interpreting that the progressive dorsoventral and, secondary, rostrocaudal development we observed are the sole organizational axes. Nevertheless, the temporal structure we observed may still closely reflect the developmental pressures that organize the vestibular ganglion. For example, the dorsal-to-ventral accumulation of ganglion neurons may be closely influenced by the spatial pressures imposed by sequential progenitor delamination: differentiated neurons are pushed dorsally by their transit-amplifying precursors in the ventral otic floor (Vemaraju et al., 2012). Similarly, though the earliest-born 30% of ganglion neurons likely receive nose-up utricular input based on their caudal position (Liu et al., 2022; Tanimoto et al., 2022), this early caudal bias is abated relatively early in time (33 hpf), with the latest-born ganglion neurons distributed evenly across the ventrorostral and ventrocaudal ganglion. We propose that for most ganglion neurons, birthdate and functional topography are uncoupled, and that spatial position is simply the passive byproduct of time.

Together, our considerations motivate future recordings in birthdated neurons to directly correlate the spatiotemporal patterns we observe with the anatomical and functional geometry of the vestibular ganglion. Such experiments may be achievable with state-of-the-art advances that overcome the incongruent axes needed to visualize the vestibular ganglion (horizontal) and stimulate the utricular macula (vertical, gravitational) (Tanimoto et al., 2022; Favre-Bulle et al., 2018; Migault et al., 2018; Sugioka et al., 2023), but which are outside the capacities of the present study. Previous work parsed the statoacoustic ganglion into coarse anteroposterior (rostrocaudal) domains by sensory input modality (auditory versus vestibular) (Vemaraju et al., 2012; Sapède and Pujades, 2010), providing a broad framework to discover corresponding anteroposterior patterns of progenitor delamination (Dyballa et al., 2017; Koundakjian et al., 2007; Bell et al., 2008; Bañón and Alsina, 2023; Vemaraju et al., 2012; Saini and Ladher, 2025), molecular differentiators (Laureano et al., 2022; Sapède and Pujades, 2010; Hammond and Whitfield, 2011; Maier and Whitfield, 2014; Whitfield et al., 2002; Haddon et al., 1998) and VIIIth nerve organization by sensory ganglia origin (Zecca et al., 2015; Kimmel et al., 1990; Pujol-Martí et al., 2010; Pujol-Martí et al., 2012). Here, the temporal organization we uncovered may provide molecular access to even finer subtypes of ganglion neurons, such as by distinct input channels or functional dynamics

(Liu et al., 2022; Liu and Bagnall, 2023) – of broad importance given the general susceptibility of the vestibular periphery in disease (Delmaghani and El-Amraoui, 2022).

## Fate and connectivity in the vestibulo-ocular reflex circuit: destined by time?

How might structural connectivity in the vestibulo-ocular reflex circuit emerge in time? Our model predicts that time reduces the complex task of matching synaptic partners across the peripheral and central nervous system to the passive assembly of first-available neurons, aligned across a common positional template. Key support comes from our finding that, as in other species (Altman and Bayer, 1981; Glover, 2003; Maklad and Fritzsch, 2003; Auclair et al., 1999; Shaw and Alley, 1981; Puelles and Privat, 1977; Vemaraju et al., 2012), across the zebrafish vestibulo-ocular reflex circuit, dorsal-to-ventral neurogenesis (Greaney et al., 2017; Goldblatt et al., 2023) proceeds over a shared temporal window (24-48 hpf) (Vemaraju et al., 2012; Greaney et al., 2017; Goldblatt et al., 2023). In spinal reflex circuits (Sürmeli et al., 2011; Balaskas et al., 2020; Kishore et al., 2020), such parallel neurogenesis imposes temporal and/or spatial constraints on axon targeting. Unlike in these circuits, sensory and central partner populations in the vestibulo-ocular reflex circuit derive from different embryological precursors with distinct environmental pressures (Haddon and Lewis, 1996), posing additional complexity to partner matching. Previously, we established that early-born, dorsally-located central vestibular neurons in the hindbrain preferentially wire to early-born, dorsal motor neurons (Goldblatt et al., 2023). Here, we demonstrate that the same temporal rules assemble vestibular ganglion neurons with their central targets. Our findings support past proposals that neuronal position – a consequence of birthdate – pre-configures connectivity in the vestibulo-ocular reflex circuit (Glover, 1996), as already established for other peripheral sensory ganglia (Pujol-Martí et al., 2010) and their central targeting (Pujol-Martí et al., 2012). Vestibular ganglion axons form a 90° turn as they enter the hindbrain and project towards central targets (Liu et al., 2022; Jia and Bagnall, 2022); this inversion could set up the dorsoventral functional axis downstream (Goldblatt et al., 2023). A 'first-come, first-served' mechanism to match temporally-available partners offers a simple means of polysynaptic sensorimotor circuit construction.

Past work suggests that the true 'first arriving' partners in the vestibulo-ocular reflex circuit are their peripheral sensory inputs. In zebrafish, hair cells in the utricular periphery become anatomically mature (Whitfield et al., 2002) and mechanically active (Tanimoto et al., 2009; Tanimoto et al., 2011) as neurogenesis is only just beginning in central/motor neurons (Greaney et al., 2017; Goldblatt et al., 2023). Similar timelines have, again, been reported in zebrafish (Haddon and Lewis, 1996; Riley et al., 1997; Vemaraju et al., 2012; Zecca et al., 2015), avians (Fritzsch and Nichols, 1993) and mammals (Altman and Bayer, 1982). Together, these findings support emerging evidence (Greaney et al., 2017; Clark et al., 2013; Goldblatt et al., 2023, 2024) that the initial synaptic architecture of the vestibulo-ocular reflex circuit develops in the anterograde (sensory-to-motor) direction. Peripheral targets can influence the development of proprioceptive spinal pathways in a similar manner (Wenner and Frank, 1995; Poliak et al., 2016; Arber, 2012; Wu et al., 2019; Norovich et al., 2022 preprint). Curiously, though central axons project to motor targets soon after neurogenesis, synaptogenesis between central and motor neurons (Goldblatt et al., 2023) and between motor neurons and muscles (Clark et al., 2013) is delayed until after differentiation across all vestibulo-ocular reflex populations, and hair cell transcriptional maturation (Baeza-Loya

et al., 2025 preprint) is nearly complete. We propose that the early formation of sensory-central connections instructs downstream wiring specificity. Evidence that vestibular ganglion axons project soon after differentiation, and that sensory-central synaptogenesis proceeds during the downstream synaptogenesis 'delay' would strongly support an anterograde assembly model of vestibulo-ocular circuit development.

Although our model predicts that temporal availability allows early-to-early born cohorts to assemble, additional forces may still refine connectivity for late-born cohorts and as circuit function matures (Leary et al., 2025). Though early input to the vestibulo-ocular reflex is exclusively nose-up (dorsocaudal), input after 30 hpf contains both up (caudal) and down (rostral) channels (Liu et al., 2022; Tanimoto et al., 2022) that may project to similar central targets, particularly to ventral neurons. This mismatch is curious, given that central projection neurons do not show strict up/down selectivity until later (Leary et al., 2025). We hypothesize that ventral central projection neurons may initially receive input from both directionally-tuned channels during development, and inappropriate sensory inputs are pruned at their central targets to enable directional selectivity in maturity. Thus, our 'first-come, first-served' model is consistent with the congruent rostrocaudal (up/down) development of the ventral vestibular ganglion. Notably, sensation is dispensable for vestibulo-ocular reflex circuit development (Ulrich et al., 2014; Roberts et al., 2017; Leary et al., 2025), arguing against stimulus-driven remodeling (Hubel and Wiesel, 1964; Schmidt and Eisele, 1985) and in favor of molecular refinement (Sanes and Zipursky, 2020); Teneurin signaling has recently been shown to correct axon mis-sorting in such a manner (Spead et al., 2023). Looking ahead, measuring when and how directional selectivity emerges in central neurons will be key to resolving the mechanisms that instantiate synaptic specificity.

### Temporal maps: a blueprint for neural circuit construction

Our findings link time with functional circuit assembly by discovering that time can shape spatial maps over relatively homogeneous functional scales, and across an entire polysynaptic sensorimotor reflexive circuit. Across vertebrate and invertebrate models, the timing of neurogenesis regulates partner availability by constraining anatomical connectivity (Berry et al., 1964; Bayraktar and Doe, 2013; Li et al., 2013; Glenwinkel et al., 2021; Koundakjian et al., 2007; Sullivan et al., 2019; Meng et al., 2020; Goldblatt et al., 2023), function in local ensembles (Fore et al., 2020; Huszár et al., 2022; Younes et al., 2024 preprint) and assembly into behaviorally-separable circuits (Wreden et al., 2017; Mark et al., 2021; McLean and Fetcho, 2009; Pujol-Martí et al., 2012; Pujala and Koyama, 2019; Hamid et al., 2024; Kishore et al., 2020; Liu et al., 2022; Fetcho and McLean, 2010; McLean et al., 2007). Here, we observed that time can also organize closely-related, binary subtypes of neurons (nose-up/nose-down) that subserve just one sensory modality (balance) and motor behavior (eye movements). Despite the clear structural constraint that time imposes on developing sensorimotor circuits, the extent to which individual systems may rely on passive 'first-come, first-served' partner matching remains controversial (Barsh et al., 2017; Isabella et al., 2021). Indeed, the next frontier in progressing beyond well-established but still indirect relationships between birth order, progressive spatial development and functional circuit assembly continues to be limited by challenges in manipulating time itself. In other systems, transplantation experiments (Barsh et al., 2017; Isabella et al., 2021) or genetic manipulations of temporally-restricted populations (Hamid et al., 2024; Sagner et al., 2021; Roome et al., 2026) have laid a path towards directly linking time, neuronal

fate and topography. For the vestibular periphery, expanded genetic atlases will be necessary for access to spatially- or temporally-restricted subtypes within an individual sensory ganglion and/or their progenitors, though the quick regenerative capacity of the zebrafish poses continuing constraints. Such insight may ultimately reconcile whether one rule, such as 'first-come first-served', or disparate rules may govern sensorimotor circuit assembly.

The continuity of the temporal map we discovered, from the sensory periphery to motor output, offers a path to resolve the extent to which the vestibulo-ocular reflex circuit is constructed holistically through shared codes or instead constructed as local modules (Cang et al., 2008), stitched together by local signals. One hypothesis is that shared organization, whether in space or time, reflects the existence of a shared molecular code that coordinates circuit assembly. Common, spatially-restricted patterns of chemorepulsive molecules underscore partner matching across retinal circuits (Sperry, 1963; Cheng et al., 1995; Flanagan and Vanderhaeghen, 1998); can be spatially inverted across target populations (Triplett and Feldheim, 2012); and can even vary along spatial gradients, matching the graded functional tuning of synaptic partners (Dombrovski et al., 2025). However, the divergent embryological origins of each vestibulo-ocular reflex circuit populations suggests the nature of such a shared code is unlikely to be spatial, downstream of common extrinsic induction factors (Dasen, 2009). Instead, our temporal map predicts that 'molecular matching' gradients might alternatively co-emerge in time. For example, time could establish 'competency' windows in which neurons are maximally poised to respond to local wiring specificity signals (Petrovic and Hummel, 2008; Yogev and Shen, 2014; Jefferis et al., 2001; Koundakjian et al., 2007; Rees et al., 2017; Sperry, 1963; Zhang et al., 2024), perhaps downstream of temporally-restricted transcription factor expression (Hamid et al., 2024; Meng et al., 2020; Sullivan et al., 2019; Isabella et al., 2021; Joo et al., 2013; Kurmangaliyev et al., 2019). Emerging evidence supports that such temporal codes may be shared across related partner populations (Shin et al., 2020) and coarsely across brain regions (Sagner et al., 2021), though whether these could act over relatively homogeneous scales remains unresolved. Future studies could leverage the access afforded by *Tg(-6.7Tru.Hcrtr2:GAL4-VP16)* to dorsal vestibular ganglion neurons, which assemble preferentially with spatially, temporally and functionally-matched central targets, to test such a hypothesis. Early work predicted that '[T]he basic connectivity scheme of the [vestibulo-ocular reflex circuit] is set up by determinate mechanisms of cell-cell recognition' (Glover, 2003), likely downstream of early fate specification programs (Glover, 1996; Lunde et al., 2019). Looking ahead, it will be illuminating to resolve whether this is achieved through distinct ligand-receptor pairs expressed in particular temporal cohorts or through just one factor, dynamically regulated or differentially available in time.

### Conclusion

Here, we complete our characterization of shared spatial, temporal and functional features across a developing sensorimotor reflex circuit. The common temporal positioning of vestibular partner populations is reminiscent of the shared topographic maps that align sensory space/body plans with central representations in canonical (i.e. retinal, auditory, somatotopic) sensory systems. For the vestibulo-ocular reflex circuit, temporal alignment may serve as a passive constraint on functional partner matching, simplifying assembly across diverse tissues. Though additional forces may subsequently refine connectivity as the vestibulo-ocular reflex circuit matures, our findings provide a key advance towards defining

the initial structural blueprints that constrain the development of polysynaptic sensorimotor circuits.

## MATERIALS AND METHODS
### Experimental model and subjects
#### Fish care
All protocols and procedures involving zebrafish were approved by the New York University Grossman School of Medicine Institutional Animal Care & Use Committee (IACUC). All larvae were raised at 28.5°C at a density of 20-50 larvae in 25-40 ml of buffered E3 (1 mM HEPES added). Larvae used for birthdating experiments were raised in constant darkness; all other fish were raised on a standard 14 h light/10 h dark cycle. Larvae for experiments were between 1-5 dpf.

#### Transgenic lines
Experiments were conducted on the $mifta^{-/-}$ background to remove pigment. Validation of statoacoustic ganglion labeling by $Tg(-6.7Tru.Hcrtr2:GAL4-VP16)$ (Lacoste et al., 2015, Schoppik et al., 2017) was performed using $Tg(-17.6isl2b:GFP)$ (Pittman et al., 2008). All other experiments used $Tg(-6.7Tru.Hcrtr2:GAL4-VP16)$ (Lacoste et al., 2015, Schoppik et al., 2017) larvae to drive expression of the following UAS reporters: $Tg(UAS:E1b-Kaede)$ (Scott et al., 2007) for birthdating only experiments and $Tg(UAS:GCaMP6s)$ (Thiele et al., 2014); $Tg(US:E1b-Kaede)$ for acute lesion/calcium imaging experiments. Larvae used were selected for brightness of fluorescence relative to siblings. Mendelian ratios were observed, supporting that selected larvae were homozygous for a given allele.

### Methods
#### Confocal imaging
Larvae were anesthetized in 0.2 mg/ml ethyl-3-aminobenzoic acid ethyl ester (MESAB, Sigma-Aldrich, E10521) before imaging except where noted. Larvae were mounted dorsal side up (axial view) or lateral side up (sagittal view) in 2% low-melting point agarose (Thermo Fisher Scientific, 16520) in E3. Birthdating images of the vestibular ganglion were collected on a Zeiss LSM800 confocal microscope with a 20× water-immersion objective (Zeiss W Plan-Apochromat 20×/1.0). All imaging windows were 160×160 µm. Anatomy stacks spanned ∼60 µm in depth (mediolateral axis), sampled every micron. For two-photon imaging of the tangential nucleus, stimulus imaging windows were 297×86 µm. Anatomy stacks of the tangential nucleus spanned 50-60 µm, sampled every micron (dorsoventral axis). All anatomy stacks were collected between 4 and 5 dpf. Raw image stacks were analyzed using Fiji/ImageJ (Schindelin et al., 2012). For details of key resources see Table S1.

#### Optical tagging of neurons by birthdate
All experiments used larvae from $Tg(-6.7Tru.Hcrtr2:GAL4-VP16)$; $Tg(UAS:E1b-Kaede)$. Neurons were optically tagged by their time of terminal differentiation using whole-embryo Kaede photoconversions (Caron et al., 2008) as described in Goldblatt et al. (2023). Briefly, embryos were photoconverted for 5 min on a custom-built apparatus with a 405 nm LED bulb, measured at 9 mW power using a 9.5 mm probe (Thorlabs, S130C). Photoconversions were performed at experimenter-defined time points between 22 and 72 hpf. Embryos used for experiments were converted at only one time point. Larvae were subsequently raised in darkness to prevent background photoconversion until the time of imaging (5 dpf). Neurons born before the time of conversion were identified by the presence of converted (red) Kaede. For each experiment, basal conversion was estimated by imaging control larvae raised in darkness until confocal imaging. $n$=5 hemispheres from $N$=5 separate larvae were analyzed for each time point. The 60 hpf time point was only analyzed with respect to 'born after' neurons. Only one hemisphere was analyzed for each larva.

#### Representations of vestibular ganglion neurons in example images and maps
The following image adjustments were performed to representative images and spatial registration data. First, to correlate the anatomy of mapped images with previous representations of vestibular ganglion topography

(Liu et al., 2022), spatial maps were rotated 3.81° anticlockwise. Next, the midpoints in the dorsoventral ($y$) and rostrocaudal ($x$) axes and circumscribed boundaries shown in spatial maps and quantification plots were defined using the mean range of the $n$=345 neurons from $N$=5 $Tg(-6.7Tru.Hcrtr2:VP16-GAL4)$ mapped control larvae. All representative images of photoconverted neurons ('born before') are shown after applying a threshold to subtract background and/or bleedthrough conversion (e.g. Figs 2B and 5E). The thresholding method is described below ('Correction of background and/or bleedthrough Kaede conversion'). In image stacks, medial cells often appeared to be dimmer due to the light scattering properties of brain tissue. To aid visualization of all cells in representative maximum intensity projection images (e.g. Figs 1B, 2B and 5E), brightness, contrast and/or gamma were separately adjusted for medial and lateral planes.

#### Imaging and analysis of bipolar afferents
Bipolar afferent development was evaluated in the same larvae used for somatic birthdating experiments. As neuronal soma had brighter fluorescence than their bipolar afferents, to avoid oversaturation we acquired separate image stacks to evaluate projections to the VIIIth nerve and utricular macula.

#### Acute lesion and calcium imaging experiments
Experiments used sighted larvae from $Tg(-6.7Tru.Hcrtr2:GAL4-VP16)$; $Tg(UAS:E1b-Kaede)$;$Tg(UAS:GCaMP6s)$. Larvae were birthdated at 26 hpf using whole-embryo conversions, then imaged on a confocal microscope at 4 dpf to identify neurons born before photoconversion. At 5 dpf, acute, uni-lateral lesion and calcium imaging experiments were performed using a two-photon microscope. Lesions were performed as originally validated in Schoppik et al. (2017) and used in Goldblatt et al. (2023) and Hamling et al. (2024). To visualize the statoacoustic ganglion, larvae were mounted lateral side up (sagittal view). Lesions were targeted to the soma of individual neurons born after the time of photoconversion (green, unconverted Kaede only) in one hemisphere only. Lesions were performed using a pulsed infrared laser (SpectraPhysics Spirit W) at 1040 mm (400 fs pulse duration, four pulses per cell over 10 ms) at 25-75 nJ per pulse. Anatomy stacks of the statoacoustic ganglion were acquired before and immediately after lesions to confirm the lesion extent (individual soma and secondarily degraded axon projections). Larvae were left to recover for 10 min and re-mounted axially for calcium imaging of the tangential nucleus. Calcium imaging after pitch-tilt stimulation was performed and analyzed as previously described (Hamling et al., 2023; Goldblatt et al., 2023, 2024). Briefly, tonic 19° up/down tilts were delivered to larvae using a galvanometer and calcium activity was imaged immediately following tilt presentation. Control experiments used non-lesioned larvae to minimize potential off-target effects from commissural-projecting neurons.

#### Acute lesion and eye tilt behavior experiments
Lesions were performed as described above. Larvae were from $Tg(-6.7Tru.Hcrtr2:GAL4-VP16)$;$Tg(UAS:E1b-Kaede)$. Larvae were birthdated at 26 hpf using whole-embryo conversions. At 5 dpf, bi-lateral lesions were performed: late-born (green, unconverted Kaede only) vestibular ganglion neurons or the anterior lateral line ganglion neurons (sham lesion control). Larvae were left to recover for 10 min before eye tilt behavior experiments.

Torsional eye rotations were measured in response to step tilts delivered using an apparatus similar in design to Bianco et al. (2012). All experiments took place in complete darkness. Lesioned larvae (vestibular ganglion or sham) were immobilized completely in 2% low-melting temperature agarose and the left eye freed. The agarose was then pinned (0.1 mm stainless minutien pins, FST) to a 5 mm² piece of Sylgard 184 (Dow Corning) which was itself pinned to Sylgard 184 at the bottom of a 10 mm² optical glass cuvette (Azzota). The cuvette was filled with ∼1 ml of E3 and placed in a custom holder on a five-axis ($x$, $y$, $z$, pitch, roll) manipulator (ThorLabs MT3 and GN2). The fish was aligned with the optical axes of two orthogonally placed cameras such that both the left utricle and two eyes were level with the horizon (front camera) and centered about the axis of rotation. The eye-monitoring camera (Guppy Pro 2 F-031, Allied Vision Technologies) used a 5× objective (Olympus MPLN, 0.1 NA) and custom

image-forming optics to create a 100×100 pixel image of the left eye of the fish (6 µm/pixel), acquired at 200 Hz. A stepper motor (Oriental Motors AR98MA-N5-3) was used to rotate the platform holding the cameras and fish. An experiment consisted of 50 cycles of four steps each. Steps were ±15° towards and away from the horizon. Each step followed a trapezoidal velocity profile peaking at 35°/s, peak acceleration 150°/s$^2$. The platform velocity and acceleration were measured using integrated circuits (IDG500, Invensense and ADXL335, Analog Devices) mounted together on a breakout board (Sparkfun SEN-09268).

### Statistical analyses and quantification

#### Validation of vestibular ganglion labeling in *Tg(-6.7Tru.Hcrtr2:GAL4-VP16)*

For initial validation that *Tg(-6.7Tru.Hcrtr2:GAL4-VP16)* labels neurons within the anatomical constraints of the vestibular ganglion, *Tg(-6.7Tru. Hcrtr2:GAL4-VP16)* expression was compared directly against *Tg(-17. 6isl2b:GFP)* in double transgenic embryos. *Tg(-6.7Tru.Hcrtr2:GAL4-VP16)* was labeled with a red UAS reporter [photoconverted *Tg(UAS-E1b:Kaede)*] and compared with co-localized *Tg(-17.6isl2b:GFP)* expression. After validating co-localization, all subsequent quantification of vestibular ganglion density and cell position (Fig. 1D-I) was performed in singly-labeled transgenic larvae. We evaluated *n*=220 neurons from *N*=3 *Tg(-17.6isl2b:GFP)* and *n*=345 neurons from *N*=5 *Tg(-6.7Tru.Hcrtr2: GAL4-VP16)* larvae.

#### Spatial registration of vestibular ganglion soma

Neurons were manually registered to a common coordinate framework in Adobe Illustrator (2021). The maximum intensity projections (MIPs) from all imaged larvae were aligned using stereotyped and clearly-identifiable anatomical landmarks: the saccule, the utricle, the edge of the otic capsule and the VIIIth nerve branch. To control for potential bias, MIP alignment was verified by two independent observers (D.G. and S.H.). To control for differences in digital magnification across microscopes and larvae, all MIPs were uniformly scaled to a standardized resolution (5.0 pixels/µm).

Vestibular ganglion neurons labeled by *Tg(-6.7Tru.Hcrtr2:GAL4-VP16)* were ~10 µm in diameter. To simplify localization of neurons in the mediolateral (*z*) axis, each anatomical image stack was subdivided into five 12 µm thick planes (Fig. 1E,G), such that each neuron would appear in no more than two mediolateral planes. Subdivisions were assessed for consistency using anatomical landmarks. The caudal-most neurons occupied the first mediolateral plane (*z*1), and the rostral-most neurons occupied the last mediolateral plane (*z*5). The mediolateral midpoint (*z*3) was qualitatively aligned with the dorsoventral (*y*) and rostrocaudal (*x*) midpoints. Each mediolateral plane was aligned with its MIP. Neurons were manually localized to a mediolateral subdivision based on the plane in which the soma fluorescence was brightest (soma center). Reference neurons, represented as circles approximately the diameter of a neuron, were centered over the soma in the appropriate subdivision. The mediolateral subdivision and the rostrocaudal (*x*) and dorsoventral (*y*) Illustrator coordinates were manually recorded for each neuron. For standardization, *xy* coordinates were normalized by subtraction of the upper-most and left-most coordinates. Standardized coordinates were then used to recreate a spatial map of neurons imaged across all fish in Matlab. Cell counts were concurrently performed in Fiji/ImageJ (Schindelin et al., 2012) using the Cell Counter plugin.

#### Representations, alignment and analysis of electron microscopy data

EM images of the vestibular ganglion, its afferent projections and the tangential nucleus, were reconstructed using Catmaid as described in Liu et al. (2022). Catmaid reconstructions were exported as a catalog SVG and imported into Illustrator for alignment. Spatial scales were preserved during image import and alignment with reference anatomical images from our *in vivo* transgenic maps. Given the minor size distortions (~10 µm) between the EM reconstructions and our *in vivo* maps, presumably a fixation artifact, the EM images were not quantitatively aligned with transgenic maps. Instead, we generated a separate reference framework as described above. Reconstructed images of the tangential nucleus, including analyses of tangential nucleus neurons that received synaptic input from utricular-innervated ganglion neurons, was based on previously-published analyses,

available in the supplementary material (supplementary data 1 and 2) in Liu et al. (2022).

#### Spatial registration and quantification of converted bipolar afferent projections

Maximum intensity projections were aligned using anatomical landmarks (otic capsule, utricle, etc.) into a hyperstack in Fiji/ImageJ (Schindelin et al., 2012). A 15 µm or 35 µm-sized rectangular region of interest (ROI) was drawn around the extent of the macula and VIIIth nerve axon projections, respectively. The fluorescence gray value along the rostrocaudal axis of the ROI was extracted using the 'Plot XY Profile' function for each channel (unconverted green, converted red), and the X profile (rostrocaudal axis) was obtained. As neurons continue to generate new, unconverted Kaede fluorescence after the time of conversion, unconverted (green) fluorescence comprises projections from neurons born both before and after the time of conversion. Fluorescence values were imported into Matlab. To account for variation in Kaede fluorescence across larvae due to transgenic expression, fluorescence values were standardized by the maximum raw fluorescence observed for each fish. Probability distributions were generated, and the median/standard deviation were calculated. *N*=3 hemispheres/time point (*N*=1 hemisphere per larva) were quantified.

#### Correction of background and/or bleedthrough Kaede conversion

Basal Kaede conversion was estimated by imaging a total of *N*=18 unconverted (green only, sibling controls) larvae raised in darkness until imaging at 5 dpf. Control larvae had comparable fluorescence as experimental larvae and were imaged alongside experimental larvae using identical imaging settings. The amount of fluorescence observed in the red (converted Kaede) channel was measured using Fiji/ImageJ (Schindelin et al., 2012). As controls were never exposed to light until the time of imaging, any red fluorescence was either basal conversion or unconverted (green) fluorescence bleedthrough. The average red Kaede fluorescence value across the vestibular ganglion was used as a 'threshold' and subtracted from all birthdated images. This ensured that cells designated as born based on the presence of red Kaede were not artifacts of basal or bleedthrough fluorescence.

#### Statistical analysis of spatial organization

For each time point, data from all hemispheres analyzed (*N*=5/time point) were pooled given variability in the number of vestibular ganglion neurons labeled across transgenic larvae (median: 69±15 neurons/fish across *N*=5 control hemispheres; see Fig. 1I). The number of 'born before' and 'born after' neurons mapped per fish is listed in Table 1. Probability distributions of soma position were generated using the mean and standard deviation from pooled and bootstrapped data (*n*=100 iterations). Spatial organization was evaluated individually in each spatial axis using a two-tailed Kolmogorov–Smirnov (KS) test given that most distributions were bimodal. All control distributions used the *n*=345 neurons within the boundaries of the vestibular ganglion mapped from the *N*=5 *Tg(-6.7Tru.Hcrtr2:GAL4-VP16)* control larvae described above.

#### Quantification and analysis of acute lesion and calcium imaging experiments

Calcium responses in the tangential nucleus were extracted and analyzed as previously described (Goldblatt et al., 2023, 2024). In Fiji/ImageJ, ROI were drawn around each tangential nucleus soma and raw fluorescence values were extracted using Matlab. Calcium responses were normalized using the mean fluorescence value of a baseline period, and the first second of the calcium response was used for analysis. Responses were analyzed with respect to two primary metrics: (1) nose-up/nose-down identity, defined using a directionality index that compared the calcium responses to up/down tilt stimulation; (2) spatial organization, mapped in Illustrator using the same methodology as described above and in Goldblatt et al. (2023, 2024). In cell position maps, plotted neuron centers were scaled to match the strength of their fluorescence responses, evaluated using the maximum value of the Δ*F*/*F* fluorescent response.

All identifiable neurons within the tangential nucleus were analyzed. We analyzed the responses of *n*=140 tangential nucleus neurons from *N*=4 control, never-lesioned larvae (median: 36±7 neurons/larva) and *n*=110

neurons from *N*=4 larvae following acute, uni-lateral vestibular ganglion lesion (median: 28±5 neurons/larva). Qualitatively, we observed that tangential nucleus neurons in larvae after vestibular ganglion lesions had lower basal fluorescence, which may account for variation in the number of identifiable neurons per larva. All Wilcoxon rank sum tests for differences of medians were performed on a per-larva basis. Statistical analyses of neuron position were performed on aggregate data from all larvae to account for spatial biases in transgenic expression.

## Analysis of eye tilt behavior data

The image was processed online by custom pattern matching software to derive an estimate of torsional angle (LabView 2014, National Instruments) and data were analyzed using custom MATLAB scripts (Mathworks). The eye's response across the experiment was first centered to remove any offset introduced by the pattern-matching algorithm. Data were then interpolated with a cubic spline to correct for occasional transient slowdowns (i.e. missed frames). The eye's velocity was estimated by differentiating the position trace; high-frequency noise was minimized using a four-pole low-pass Butterworth filter (cutoff=3 Hz). Each step response was evaluated manually; trials with rapid deviations in eye position indicative of horizontal saccades or gross failure of the pattern matching algorithm were excluded from analysis. The response to each step for a given fish was defined as the mean across all responses to that step across cycles. The gain was estimated by measuring the peak eye velocity occurring over the first second after the start of the step.

## Statement on artificial intelligence use

No artificial intelligence tools were used in the preparation of this manuscript, including code generation, image production, data collection, analysis, and writing.

## Acknowledgements

The authors thank Hannah Gelnaw for assistance with fish care, Martha Bagnall for invaluable insights and assistance with EM data, and Jeremy Dasen, Claude Desplan, Katherine Nagel, Dan Sanes and Masashi Tanimoto along with the members of the Schoppik and Nagel labs for their valuable feedback and discussions. This research was supported by the Intramural Research Program of the National Institutes of Health (NIH), National Institute of Neurological Disorders and Stroke (NINDS). The contributions of the NIH author(s) were made as part of their official duties as NIH federal employees, are in compliance with agency policy requirements, and are considered Works of the United States Government. However, the findings and conclusions presented in this paper are those of the author(s) and do not necessarily reflect the views of the NIH or the U.S. Department of Health and Human Services.

## Competing interests

The authors declare no competing or financial interests.

## Author contributions

Conceptualization: S.H., D.S., D.G.; Formal analysis: S.H., E.G.; Funding acquisition: D.S., D.G.; Investigation: S.H., E.G., M.R.G., S.N.D.; Methodology: S.H., M.R.G., S.N.D., D.S., D.G.; Supervision: D.S., D.G.; Validation: S.H.; Visualization: S.H.; Writing – original draft: S.H.; Writing – review & editing: D.S., D.G.

## Funding

Research was supported by the National Institute on Deafness and Communication Disorders of the National Institutes of Health under award number R01DC017489 and the National Institute of Neurological Disorders and Stroke under award numbers F99NS129179 and T32NS086750. Open Access funding provided by the National Institutes of Health. Deposited in PMC for immediate release.

## Data and resource availability

For further information and requests for resources and reagents contact D.G. and D.S. All data and code are deposited at the Open Science Framework and are publicly available at https://doi.org/10.17605/OSF.IO/KP839. All other relevant data and details of resources can be found within the article and its supplementary information.

## The people behind the papers

This article has an associated 'The people behind the papers' interview with some of the authors.

## Peer review history

The peer review history is available online at https://journals.biologists.com/dev/lookup/doi/10.1242/dev.204616.reviewer-comments.pdf

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
