## [Peer Review File · Development (Cambridge, England)]

Birthdate aligns vestibular sensory neurons with central and motor partners across a sensorimotor reflex circuit for gaze stabilization

Stephanie Huang, Emily Gershowitz, Marie R. Greaney, Samantha N. Davis, David Schoppik and Dena Goldblatt

DOI: 10.1242/dev.204616

Editor: Steve Wilson

Review timeline

Original submission:	23 December 2024
Editorial decision:	21 March 2025
First revision received:	6 December 2025
Editorial decision:	8 January 2026
Second revision received:	9 January 2026
Accepted:	12 January 2026

Original submission

First decision letter

MS ID#: dev.204616

MS TITLE: Birthdate aligns vestibular sensory neurons with central and motor partners across a sensorimotor reflex circuit for gaze stabilization

AUTHORS: Stephanie Huang, Emily Gershowitz, Marie R. Greaney, Samantha N. Davis, David Schoppik and Dena Goldblatt

Dear Dena,

Many apologies for the length of time it took to obtain reviews on your manuscript. However, I have now received all the referees' reports on the above manuscript, and have reached a decision. The referees' comments are appended below, or you can access them online: please go to:

As you will see, the reviews are somewhat mixed with referee two less enthusiastic than the others. However, there are some shared opinions between reviews and all three have suggestions to address their concerns and improve the manuscript. If you are able to revise the manuscript along the lines suggested, I will be happy to receive a revised version of the manuscript. Please also note that Development will normally permit only one round of major revision. If it would be helpful, you are welcome to contact us to discuss your revision in greater detail. Please send us a point-by-point response indicating your plans for addressing the referees' comments, and we will look over this and provide further guidance.

Please attend to all of the reviewers' comments and ensure that you clearly highlight all changes made in the revised manuscript. Please avoid using 'Tracked changes' in Word files as these are lost in PDF conversion. I should be grateful if you would also provide a point-by-point response detailing how you have dealt with the points raised by the reviewers in the 'Response to Reviewers' box. If you do not agree with any of their criticisms or suggestions please explain clearly why this is so.

Reviewer 1

Advance summary and potential significance to field

First, the authors showed that the Tg(-6.7Tru.Hcrtr2:GAL4-VP16) line provides reliable access to the vestibular ganglion neurons (VGN) that were previously described using the Tg(-17.6isl2b:GFP) fish line, and confirmed that these neurons are located in roughly the same place in the hindbrain. To tag vestibular ganglion neurons during development, researchers used a photoconvertible reporter, kaede, to label neurons born at different stages, revealing that early-born neurons are organized dorsally and late-born neurons ventrally, indicating a temporal and spatial pattern in neuron development. They also showed that bipolar afferents of VGN and the VIIIth nerve ending are organized in a way where early-born neurons project caudally in comparison to late-born neurons. Finally, the authors showed that early-born vestibular ganglion neurons connect with spatiotemporally and functionally matched central neurons, forming specific connections based on their birthdate and position.

I am excited about these results, which further strengthen the critical link between the birthdates of neurons and their function/connectivity within brain circuits. The authors have presented a well-conducted study, with beautiful and clear figures to communicate their message. I particularly liked the summary scheme at the end, which put together the authors' main findings. I have a few comments, which I hope might help the authors clarify their points and strengthen their message. I support the publication of this paper in Development.

Comments:

1) In general, I find that the text is rather heavy to read. As a person not working on the vestibular system and as a non-native speaker, I would have appreciated a clearer text with shorter sentences. Also, the text font that the manuscript is delivered in was particularly difficult to read, which might have contributed to me reading the text in multiple repeats at times. I truly hope the authors will pay attention to this in the next revision.

2) The authors used laser ablation for testing their hypothesis on the connectivity of neurons born and projecting to different regions within the vestibular circuit. While the specificity of these ablations is a matter to consider, I wonder if some additional points can be made with respect to this critical manipulation. For example, if the location of cell bodies and projections is stereotypical across animals, one might aim for cell body ablations too. I am aware that this might be a time-consuming experiment, and I do not demand this experiment. But I think the authors should at least clarify these ablations better.

3) Also, ablation experiments clearly show that a functionally distinct component of the circuit is now perturbed without perturbing other components (tested by calcium imaging). This was really an ingenious experiment, which I liked a lot. I assume these results have a direct prediction that some ocular reflexes will be perturbed and some not. Would this hypothesis be testable using behavioral recordings? Or perhaps laser ablation weakens the animals too much to perform such experiments.

4) Why is the ratio of overlap between Tg(6.7Tru.Hcrtr2:GAL4-VP16) and Tg(-17.6isl2b:GFP) neurons' somata not reported, but primarily focused on spatial overlap?

Reviewer 2*Advance summary and potential significance to field*

Huang and colleagues present experiments to birthdate vestibular ganglion neurons and compare to function in forming central connections. The authors use kaede photoconversion to birthdate vestibular ganglion neurons providing evidence for spatial differences within the ganglion corresponding to order of birth. They also provide evidence that axon projections both peripherally and centrally have a spatial arrangement corresponding to birth order. They then use ablation of a subset of neurons in the ganglion to relate these spatial maps to functional activity in central targets of the vestibular ganglion. Together with previous studies they suggest that birth order is the key feature in establishing a functional circuit for vestibular driven behavior. While the quantitative analysis of vestibular ganglion birthdate is reasonably done, the evidence for specific circuit assembly is very indirect and does not have the resolution to address the question of whether central and peripheral neuron birth date factors into functional wiring.

Comments for the author

I find the framing of the development of the ganglion as specifically along the dorsoventral axis as confusing. The authors present data showing there is also a significant rostrocaudal bias found in the earliest born neurons (before 26h and after 30h, Fig 3 and Table 2). Moreover the analysis of the axonal projections suggests organization is also along a rostrocaudal axis. These are consistent with the previous functional analysis of Tanimoto et al 2022. (The Tanimoto et al. 2022 referenced as a preprint is now published: Nat Commun. 2022 Dec 21;13:7622 doi: 10.1038/s41467-022-35190-9.)

The authors use an ablation experiment to support a functional dorsoventral axis in the ganglion. However, the functional data presented here are only indirectly reflect ganglion activity, measuring central activity after ganglion ablation. They appear to directly contradict the clear rostrocaudal functional axis demonstrated by direct imaging of the ganglion by Tanimoto et al. Part of the confusion may be what neurons were ablated and which were spared. If I understand correctly only neurons born before 26h were spared. If this is the case then those in the dorso-caudal sector were left intact and dorso-rostral and all ventral neurons removed. Based on Tanimoto et al's analysis, there would be a loss of all rostral (nose down) response and partial loss of caudal (nose up) response. Indeed there appears to be some loss of nose up response in Fig 5F as well as loss of nose down response in 5G. In any event it appears that a case for simple dorsoventral organization as critical for circuit development is an oversimplification. Moreover it is therefore not clear to what degree the connections between ganglion and central targets in fact corresponds to birth order or anticipate functional sensory-to-central partner matching. It may be that many late born neurons in the ganglion connect to late born neurons in the hindbrain. However if these ventral neurons respond to both nose up and nose down tilts then how does birth order relate to the function of the circuit?

The authors note that they were only able to perform ablation of a subset of neurons at 5 dpf due to spatial constraints. Additional experiments ablating neurons at earlier times would potentially solve these problems and provide additional tests of the relationships between birthdate and functional connection, including ideas about "first-come, first-served".

It should be noted that the idea that birth order controls central targeting of peripheral sensory neurons is not particularly new - e.g. was previously demonstrated for somatotopy of lateral line ganglion neurons specifically targeting distinct regions in in the zebrafish hindbrain was previously demonstrated by Pujol-Marti and colleagues (doi: 10.1002/dvdy.22320; doi: 10.1523/JNEUROSCI.5157-11.2012.).

Reviewer 3

Advance summary and potential significance to field

This paper by Huang and colleagues examines the development of the statoacoustic ganglion and provides convincing evidence that the birth date and position of SAG cells correlates with their functional tuning and their connectivity to central and peripheral targets. Along with previous work from this group and others, this supports an elegant model in which birth order and topography likely shape the construction of an entire, evolutionarily conserved, sensorimotor pathway. The data and analyses are generally very good, with one exception, discussed below. A highlight of the paper was a clever experiment combining Kaede birthdating, "TIPM" calcium imaging, and laser-ablations that was used to show that sensory and central vestibular cells form functional circuits based on birth order and anatomical location. I expect the paper will certainly be of interest to the readership of Development.

Figure 1 makes the case for using a *hcrtr2* line to look at SAG neurons. The emphasis is on soma locations but this alone might not be strong evidence (for example if SAG neurons were intermingled with another cell type). Later in the paper, Kaede conversions show the *hcrtr2* cells project axons to the utricular macula and in the VIIIth nerve – why not include some of this anatomy in F1 to show the neurites go to the right places?

My main concern with the paper related to figure 3 and the inferences about dorsoventral versus rostrocaudal developmental gradients. The data seems solid enough, but the results text (around Ln93 onwards) and logic of the argument is very difficult to follow. As far as I could understand the results, the authors observe early born cells in dorsal and caudal SAG (e.g. 22 and 26 hpf 'born before' cells in F3A). After that stage, cells seem to appear both rostrally and ventrally. Yet the authors somehow conclude that only DV development is of concern. ie "Dorsoventral, but not rostrocaudal development, is the primary predictor of late-born somatic accumulation..." and "Unlike its rostro-caudal functional organization, we discovered that the development of the vestibular ganglion proceeds along a dorsal-to-ventral axis." Perhaps a source of confusion relates to when the threshold is for describing cells as late-born? Whatever case, the data do clearly show there is a topography related to birth order in the SAG in which the earliest cells are localised dorso-caudally: Perhaps F6B might better have the pink cells confined to such a quadrant?

Minor:

Tanimoto et al 2022 is now published, Is there a reason to cite the biorXiv preprint (ref 24)?

F1A: Inset also shows fluorescence micrograph (to visualise Kaede)?

F1B: dashed circle is black.

F1D: for the 220 neurons shown - presumably randomly selected?

In Table 1 and F1D its not very clear what is meant by "control". All transgenically labelled cells as an estimate of total SAG population?

Similarly "The early-born cohort was caudal of center relative to control (pKS test=1.0x10⁻¹⁰).

Mediolateral organization was not observed (pKS test=0.25)":

Again, not clear what is meant by control. Is the mediolateral KS test a comparison of the early vs late distributions, or something else?

F3C-D y-axis need units

At Ln99, the term "undifferentiated" neurons is introduced. What does this refer to?

F4B: Dashed and dotted appear the same.

F4D-E: Suggest rewriting the legend text to state explicitly what is actually being imaged in D. The cartoon suggests an axon tract en route to the macula yet the results text implies it is the macula.

Ln136: "The matched connectivity..." Connectivity is not being assessed.

In the key resources table and transgenic lines section of methods, please try to report the allele numbers for mutations and transgenes. I believe the table should list the genomic features for extant lines (of the form ZDB-ALT-...), not the construct identifier.

Ln313: A measure of irradiance from the 405nm LED would make it easier to reproduce the photoconversions.

First revision

Author response to reviewers' comments

We appreciate the Reviewers' support for our manuscript and their patience, as a catastrophic fire in our fish facility (>90% colony loss) delayed our resubmission. We've carefully considered all of the Reviewers' suggestions and have made several improvements, including

(i) a new major analysis of a previously-published EM connectome of the vestibular ganglion, bolstering our transgenic analyses in nearly all figures, (ii) an expanded Results section that more closely evaluates our findings relative to this and other published works; and (iii) tempered claims throughout the text about our spatial findings, particularly in the Discussion (see new “Limitations” section). We believe our manuscript is considerably stronger and are grateful to the Reviewers for their time, thoughtfulness, and encouragement.

Reviewer 1

Summary: First, the authors showed that the Tg(-6.7Tru.Hcrtr2:GAL4-VP16) line provides reliable access to the vestibular ganglion neurons (VGN) that were previously described using the Tg(-17.6isl2b:GFP) fish line, and confirmed that these neurons are located in roughly the same place in the hindbrain. To tag vestibular ganglion neurons during development, researchers used a photoconvertible reporter, kaede, to label neurons born at different stages, revealing that early-born neurons are organized dorsally and late-born neurons ventrally, indicating a temporal and spatial pattern in neuron development. They also showed that bipolar afferents of VGN and the VIIIth nerve ending are organized in a way where early-born neurons project caudally in comparison to late-born neurons. Finally, the authors showed that early-born vestibular ganglion neurons connect with spatiotemporally and functionally matched central neurons, forming specific connections based on their birthdate and position.

I am excited about these results, which further strengthen the critical link between the birthdates of neurons and their function/connectivity within brain circuits. The authors have presented a well-conducted study, with beautiful and clear figures to communicate their message. I particularly liked the summary scheme at the end, which put together the authors' main findings. I have a few comments, which I hope might help the authors clarify their points and strengthen their message. I support the publication of this paper in Development.

We appreciate the Reviewer's enthusiasm for our manuscript, their thoughtful suggestions for how to improve the clarity and presentation of the text, and their support for publication in Development. We were especially delighted by the exciting behavioral experiment the Reviewer proposed, which we were able to include in our revised manuscript.

Major comments:

1) In general, I find that the text is rather heavy to read. As a person not working on the vestibular system and as a non-native speaker, I would have appreciated a clearer text with shorter sentences. Also, the text font that the manuscript is delivered in was particularly difficult to read, which might have contributed to me reading the text in multiple repeats at times. I truly hope the authors will pay attention to this in the next revision.

We appreciate the Reviewer's concern and have given particular attention to text clarity, including sentence structure, in this revision; we believe the text is now considerably stronger for their suggestions. We have used a standard font for the revision and hope the Reviewer finds it easier to read.

2) The authors used laser ablation for testing their hypothesis on the connectivity of neurons born and projecting to different regions within the vestibular circuit. While the specificity of these ablations is a matter to consider, I wonder if some additional points can be made with respect to this critical manipulation. For example, if the location of cell bodies and projections is stereotypical across animals, one might aim for cell body ablations too. I am aware that this might be a time-consuming experiment, and I do not demand this experiment. But I think the authors should at least clarify these ablations better.

We've clarified in the Results (lines 239-243) and Methods (line 507) that our laser ablations specifically targeted the cell bodies of individual vestibular ganglion neurons, which in turn degrades their projections.

The specificity of our ablations is an important consideration. Several of our past papers have performed comparable laser ablation experiments and have validated their specificity (see: Bianco et al 2012, Schoppik et al. 2017, DOI [10.1523/JNEUROSCI.1711-17.2017](https://doi.org/10.1523/JNEUROSCI.1711-17.2017); Hamling et al. 2024, DOI [10.1523/JNEUROSCI.2315-23.2024](https://doi.org/10.1523/JNEUROSCI.2315-23.2024); and Goldblatt et al. 2023, DOI [10.1016/j.cub.2023.02.048](https://doi.org/10.1016/j.cub.2023.02.048)). We've clarified our prior use of our approach in the Results and have added citations to relevant work where appropriate (line 262).

3) Also, ablation experiments clearly show that a functionally distinct component of the circuit is now perturbed without perturbing other components (tested by calcium imaging). This was really an ingenious experiment, which I liked a lot. I assume these results have a direct prediction that some ocular reflexes will be perturbed and some not. Would this hypothesis be testable using behavioral recordings? Or perhaps laser ablation weakens the animals too much to perform such experiments. This is an excellent suggestion. We performed this experiment and have incorporated the results into Fig. 5 and the relevant Results section. Briefly, we repeated our lesion experiments and then performed an established behavioral assay to measure the velocity and magnitude of eye rotations in response to up/down table tilts (Schoppik et al. 2017, DOI [10.1523/JNEUROSCI.1711-17.2017](https://doi.org/10.1523/JNEUROSCI.1711-17.2017); Leary et al. 2025, DOI [10.1126/science.adr9982](https://doi.org/10.1126/science.adr9982)). We did in fact observe that lesion of late-born vestibular ganglion neurons (all nose-down, some nose-up) almost entirely abolished ocular responses to nose-down tilts, with moderate abatement of nose-up (eyes-down) behavior. These observations are in line with our calcium imaging data that demonstrates similar impairments in central neuron tilt responses. Full results and statistics can be found in lines 259-268, with expanded Methods included under both Method Details (“Acute lesion and eye tilt behavior experiments”) and Statistical Analyses (“Analysis of eye tilt behavior data”).

We thank the Reviewer again for their suggestion and their patience, as it took quite some time to pilot these experiments and evaluate technical feasibility, and then to build a sufficiently large N for statistical robustness. We’re truly delighted we were able to include these findings in our revised manuscript.

4) Why is the ratio of overlap between Tg(6.7Tru.Hcrtr2:GAL4-VP16) and Tg(-17.6isl2b:GFP) neurons' somata not reported, but primarily focused on spatial overlap?

We take the Reviewer’s point that a direct ratio of overlap can provide complementary validation. Initially, to validate that Tg(6.7Tru.Hcrtr2:GAL4-VP16)-labeled vestibular ganglion neurons co-localize with Tg(-17.6isl2b:GFP), we evaluated co-expressing neurons from the same larvae. However, all quantification was performed in **separate** animals (i.e., expressing either Tg(6.7Tru.Hcrtr2:GAL4-VP16) or Tg(-17.6isl2b:GFP), but not both). This is because we found that Tg(6.7Tru.Hcrtr2:GAL4-VP16) *heterozygotes* have more variable expression, presumably due to the GAL4/UAS expression system, (extensive prior work using this transgenic line in Goldblatt et al. 2023, DOI [10.1016/j.cub.2023.02.048](https://doi.org/10.1016/j.cub.2023.02.048)). Therefore a same-animal quantification of neurons that expressed both Tg(6.7Tru.Hcrtr2:GAL4-VP16) and Tg(-17.6isl2b:GFP) would be too noisy to be generalizable. Instead, we believe our quantification of the spatial overlap between these populations is sufficient to demonstrate the reliability of Tg(6.7Tru.Hcrtr2:GAL4-VP16) in labelling vestibular ganglion neurons. We’ve clarified the Methods (“Validation of vestibular ganglion labeling”) and appropriate figure captions (e.g., Fig. 1D) to reflect that all quantification was performed in separate transgenic larvae.

We note that doubly biallelic larvae used for later birthdating experiments have consistent expression. While it is technically possible to redo all the quantification experiments on this background, this would require making fish with at least 7 alleles (nacre -/-, GAL4 +/-; UAS +/-; -17.6isl2b:GFP +/-) that would take multiple generations. We do not believe that there is enough additional information in the direct ratio measurement to merit the considerable effort to perform this experiment.

Reviewer 2

Summary: Huang and colleagues present experiments to birthdate vestibular ganglion neurons and compare to function in forming central connections. The authors use kaede photoconversion to birthdate vestibular ganglion neurons providing evidence for spatial differences within the ganglion corresponding to order of birth. They also provide evidence that axon projections both peripherally and centrally have a spatial arrangement corresponding to birth order. They then use ablation of a subset of neurons in the ganglion to relate these spatial maps to functional activity in central targets of the vestibular ganglion. Together with previous studies they suggest that birth order is the key feature in establishing a functional circuit for vestibular driven behavior. While the quantitative analysis of vestibular ganglion birthdate is reasonably done, the evidence for specific circuit assembly is very indirect and does not have the resolution to address the question of whether central and peripheral neuron birth date factors into functional wiring.

We appreciate the Reviewer’s thoughtful examination of our manuscript. We’ve considerably expanded the Results and Discussion to explicitly contextualize our findings with prior connectomics

findings (Liu et al. 2022), functional imaging (Tanimoto et al. 2022), and other foundational work in the sensory periphery (Pujol-Marti et al. 2010 & 2012). The expanded Results enabled a more close Discussion of where our findings directly support and extend these past works and potential sources of discrepancies (new “Limitations” section). Additionally, we’ve used the Discussion to reflect on the broad challenges the field continues to face in directly relating temporal and spatial development with the features of mature neurons. We hope the Reviewer agrees that our expanded Results/Discussion, as well as tempered claims throughout the text, substantially bolsters our conclusions.

Major comments:

1) I find the framing of the development of the ganglion as specifically along the dorsoventral axis as confusing. The authors present data showing there is also a significant rostrocaudal bias found in the earliest born neurons (before 26h and after 30h, Fig 3 and Table 2). Moreover the analysis of the axonal projections suggests organization is also along a rostrocaudal axis. These are consistent with the previous functional analysis of Tanimoto et al 2022.

We acknowledge the shared consideration with Reviewer 3 that our original manuscript did not sufficiently appreciate the rostrocaudal organization we observed at early developmental stages. Initially, we under-appreciated the caudal bias observed through 30 hpf given several findings: (i) that this bias persists for only 15% of the total differentiation time we measured (8/50 hours); (ii) that this represents only 30% of *hcrtr2+* neurons (Table 1), and (iii) that the caudal location of these neurons is very close to the midpoint demarcation line, raising ambiguity as to whether this is a true and enduring developmental axis. **To resolve these ambiguities, we’ve included several major analyses and new considerations in the text:**

(1) To acknowledge the challenges associated with variable transgenic labeling and spatial registration across small scales (the rostrocaudal axis spans only 55 μm), we’ve added a new figure (Fig. 2F) showing individual maps of early- and late-born neurons from the five fish birthdated at 26 hpf. These five fish are included in the aggregate map shown in Fig. 2D (N=10 larvae, comprised of N=5 fish birthdated at 22 hpf and the N=5 fish birthdated at 26 hpf shown in Fig. 2F). We believe these maps better represent the ambiguity and variability in the rostrocaudal position of these early-born neurons; for example, neurons in Fish 2, 3, and 5 fall extremely close to the spatial midpoint, while Fish 1 and Fish 4 are more caudally-positioned.

(2) Variability in genetic labeling within and across transgenic lines remains a general challenge for zebrafish studies. Therefore, we integrated a **new dataset** into Figures 1-2. Specifically, we re-analyzed the EM connectomic data previously published by Liu et al. 2022 (n=106 neurons from N=1 hemisphere), taking care to align the EM reconstructed neurons with our *in vivo* transgenic maps and carefully evaluate how well the two transgenic lines we used for our initial evaluation (*hcrtr2* and *isl2b:GFP*) compare to this “ground truth” reference atlas. A detailed evaluation can be found in Results (lines 86-105) and throughout Fig. 1.

(3) Our incorporation of EM data allowed us to perform a new **major analysis**. The prior EM study inferred neuronal age using myelination at 5 dpf. We mapped the location of myelinated (inferred to be earlier-born) and unmyelinated (late born) to our reference framework. We again find that myelinated neurons lie extremely close to the rostrocaudal midpoint (Fig. 2G-I). Again, this merits a cautious interpretation of whether the earliest-born neurons are truly caudally-localized. We note that myelination at 5 dpf is well-after the early-born subset (dorsal, potentially caudal) is born (26 hpf), and thus is likely not exclusive to this subset.

(4) Nevertheless, the caudal bias we observed has important implications given the rostrocaudal functional topography of the ganglion, as described by Tanimoto et al. 2022. To more explicitly compare our findings, we consulted with the authors; our discussion is now summarized in the results as “personal communication” with M. Tanimoto’s permission (line 109-112). Most importantly, we discovered that while our *hcrtr2* line and their *myo6b* line access similar spatial extents of the vestibular ganglion, the density of labeled neurons varies substantially: *hcrtr2+* appears to label 50-70% of the *myo6b* population. However, Tanimoto et al. also report a substantial number of immature ganglion neurons that are not tilt-responsive at 5 dpf. We’d previously noted in the Discussion that the vestibular ganglion undergoes adult neurogenesis per Schwarzer et al. 2020 (DOI [10.1242/dev.176750](https://doi.org/10.1242/dev.176750)). We interpret that additional labeling in *myo6b* could include these immature neurons, which may not be incorporated by either our *hcrtr2*

transgenic or in *Tg(isl2b:GFP)* as quantified in Fig. 1.

(5) Generally, we now acknowledge the challenges and limitations associated with variable transgenic line access to the vestibular ganglion, as well as their implications for interpreting our spatial maps, where relevant in the Results (lines 114-121; lines 179-184) and in a new Discussion section (“Limitations”; lines 289-315). Likewise, we’ve substantially tempered claims that the dorsoventral axis is the sole predictor of organization; included the rostrocaudal axis as a secondary predictor, with caveats as discussed above; and cautioned against interpretations that the spatial maps we’ve uncovered are the only organizational axis given the above considerations.

(6) The Reviewer is also correct that axonal organization was observed along the rostrocaudal axis, consistent with prior functional imaging results. We note that analysis in the dorsoventral axis is not possible here, as individual vestibular ganglion neurons inherently form bipolar projections with outgrowth in both dorsal (to central targets) and ventral (to the utricular macula) directions. We have reinforced this constraint in the Results (line 191-192).

(7) Lastly, we acknowledge that some confusion may derive from how we’ve classified neurons as “early-born” or “late-born”. We’ve clarified the Results (lines 136-139) that our designations are relative to when 50% of *hcrtr2+* neurons are born (36 hpf), rather than the true temporal midpoint of differentiation (48 hpf, the midpoint of 22-72 hpf). We hope this better clarifies that a spatial pattern in the first 8 hours of development is relatively transient, and may only explain variability for a subset (30%) of neurons.

The Tanimoto et al. 2022 referenced as a preprint is now published: Nat Commun. 2022 Dec 21;13:7622 doi: 10.1038/s41467-022-35190-9.

We thank the Reviewer for their attention to detail and have updated this reference.

2) The authors use an ablation experiment to support a functional dorsoventral axis in the ganglion. However, the functional data presented here are only indirectly reflect ganglion activity, measuring central activity after ganglion ablation. They appear to directly contradict the clear rostrocaudal functional axis demonstrated by direct imaging of the ganglion by Tanimoto et al.

We agree that the ideal experiment would directly record vestibular ganglion activity following utricular stimulation. On a static, upright microscope, the axis needed to visualize the vestibular ganglion (horizontal) is incongruent with the vertical gravitational axis necessary to stimulate the utricular macula. Current state-of-the-art technological workarounds include tiltable, rotating microscopes as in Tanimoto et al. 2022 and Migault et al. 2018 (DOI 10.1101/2024.03.22.586054) or alternative stimulation methods such as optical trapping (e.g., Favre-Bulle et al. 2017, DOI 10.1016/j.cub.2018.09.060), all of which are well beyond the current capacities of our lab. Instead, we rely on past work from our lab that establishes that indirect imaging of central vestibular neurons is a veridical readout of vestibular ganglion activity following utricular stimulation (see: Hamling et al. 2022, Goldblatt et al. 2023). We’ve expanded the Discussion to note the considerable value of these future experiments (lines 316-320).

However, we respectfully disagree with the Reviewer’s claim that our findings contradict the rostrocaudal functional axes observed by Tanimoto et al. 2022 and indirectly, by Liu et al. 2022. We point to EM reconstructions from Liu et al. 2022 of ganglion afferent projections to central hindbrain targets, which report that the afferent bundle forms a 90-degree turn at the hindbrain entry site. We predict this inversion could set up the dorsoventral functional axis we observe in the central nucleus in the present study and past work (Goldblatt et al. 2023), which is further supported by EM reconstructions of functional central topography (Liu et al. 2022). We’ve acknowledged this in the Discussion (lines 347-349).

To bolster our conclusions, we’ve now incorporated these EM reconstructions directly into Fig. 5. Again inferring neuronal age based on myelination, we find that the afferent bundle inverts at the hindbrain entry site, such that myelinated afferents (lateral as they enter the VIIIth nerve) innervate the dorso-medial tangential nucleus (Fig. 5B). We then extended these analyses to visualize ganglion afferents that form direct synaptic connections onto central neurons (Fig. 5C-D), which further support the dorsal organizational axis.

Next, we note that prior work from the Riley lab (Vemaraju et al. 2012, DOI

[10.1371/journal.pgen.1003068](https://doi.org/10.1371/journal.pgen.1003068)) demonstrated that post-mitotic ganglion neurons are pushed dorsally into the mature ganglion, a passive byproduct of sequential patterns of progenitor delamination from the ventral otic floor. Thus, our finding of dorsal development is well in line with Vemaraju et al's model and validates that development and function may very well be uncoupled. We've noted this more clearly in the Introduction (line 44-46) and Discussion (lines 309-311).

Lastly, we point to the Reviewer's next comment as validation that our findings actually **support** the major result from Tanimoto et al. 2022, rather than contradict:

Part of the confusion may be what neurons were ablated and which were spared. If I understand correctly only neurons born before 26h were spared. If this is the case then those in the dorso-caudal sector were left intact and dorso-rostral and all ventral neurons removed. Based on Tanimoto et al's analysis, there would be a loss of all rostral (nose down) response and partial loss of caudal (nose up) response. Indeed there appears to be some loss of nose up response in Fig 5F as well as loss of nose down response in 5G.

The Reviewer is correct that elimination of all neurons but the dorso-caudal sector leads to a near-complete loss of nose-down responses in the central vestibular nucleus and a partial loss of nose-up responses, consistent with the functional predictions from Tanimoto et al. 2022. Our new behavioral experiments, suggested by Reviewer 1 (ocular responses after late-born ganglion neuron lesion) further agree with these findings.

All in all, we agree with the Reviewer that a closer comparison of our findings with Tanimoto et al. 2022 is important. As described above, we've expanded the Results and Discussion to consider how the differences in these labeled populations could contribute to our findings and more explicitly describe where our results (e.g., ganglion lesion and central vestibular nucleus imaging) directly support and extend the rostrocaudal functional axis described previously.

In any event it appears that a case for simple dorsoventral organization as critical for circuit development is an oversimplification.

We agree with the Reviewer and have expanded this considerably throughout the Results and Discussion as noted above.

Moreover it is therefore not clear to what degree the connections between ganglion and central targets in fact corresponds to birth order or anticipate functional sensory-to-central partner matching. It may be that many late born neurons in the ganglion connect to late born neurons in the hindbrain. However if these ventral neurons respond to both nose up and nose down tilts then how does birth order relate to the function of the circuit?

The coincidental emergence of ventro-caudal (nose-up) and ventro-rostral (nose-down) neurons in our dataset was an especially interesting finding, given our observation that early-born (dorsal) neurons preferentially assemble with central neurons of the same type. However, we do not think it is *incongruent* with our model. In our original Discussion, we considered that vestibular ganglion afferents may in fact project and assemble with temporally-matched (dorsal to ventral targets), and additional molecular forces subsequently prune and/or refine connectivity in time. Unpublished observations from our lab support that the selective directional tuning (up- or down-only) of central neurons is in fact a feature of late development, rather than immediately conveyed; this further supports a role for later molecular refinement of connectivity. We've expanded the Discussion with specific hypotheses for how this may arise (lines 370-379).

3) The authors note that they were only able to perform ablation of a subset of neurons at 5 dpf due to spatial constraints. Additional experiments ablating neurons at earlier times would potentially solve these problems and provide additional tests of the relationships between birthdate and functional connection, including ideas about "first-come, first-served".

We agree with the Reviewer that our ablation experiments are limited by our choice of acute 5 dpf lesions. We've extensively explored technical paths for earlier lesions with, unfortunately, no clear path forward. For example, we've attempted to lesion vestibular ganglion neurons during early development (24-30 hpf) - and have been thwarted by the rapid (24-48 hours) regeneration of the zebrafish ganglion and continued replenishment of differentiated neurons by their progenitor pool (see: Vemaraju et al. 2012, DOI [10.1371/journal.pgen.1003068](https://doi.org/10.1371/journal.pgen.1003068); Schwarzer et al. 2020, DOI [10.1242/dev.176750](https://doi.org/10.1242/dev.176750)). To attempt to circumvent the latter, we ablated their

sox10+ progenitor pool (16-20 hpf), again without success. Theoretically, one could alternatively attempt to manipulate early-born ganglion neurons through genetic ablation - but spatially-restricted markers within the vestibular ganglion have not yet been identified. Lastly, while transplantation of early-born (dorsal) neurons to ventral locations would speak to whether time does instruct connectivity (see: Barsh et al. 2017, DOI [10.1016/j.cub.2017.11.022](https://doi.org/10.1016/j.cub.2017.11.022)), this approach is beyond our current capabilities. We now discuss these limitations (lines 392-400).

Adding to the complexity of such an experiment: our lab recently demonstrated that downstream central neurons do not acquire their stable up/down tuning profiles until approximately (Leary et al. 2025, DOI: [10.1126/science.adr9982](https://doi.org/10.1126/science.adr9982)), well-after spatially-restricted development is discontinued in the vestibular ganglion. Thus, even if an acute lesion earlier in development was possible, it would not yield easily-interpretable results.

While the Reviewer is correct that our indirect approach is not without caveats, we do feel the preponderance of evidence - that is, the orderly temporal and spatial matching of developing vestibular ganglion neurons and their central targets, in addition to the functional connection at maturity between spatiotemporally-matched populations *and* our new incorporation of EM data supporting these connectivity patterns - supports our claims that the VOR is assembled in a “first-come, first-served” manner.

Lastly, we acknowledge an important point the Reviewer raises about the broad challenge of directly testing the role of time in circuit formation. To our knowledge, the only studies to have successfully done so have relied on genetic manipulations of early-born neurons (e.g., in mammals, Sagner et al. 2021, DOI [10.1371/journal.pbio.3001450](https://doi.org/10.1371/journal.pbio.3001450) and more recently, Roome et al. 2025, DOI [10.1101/2025.03.14.643370](https://doi.org/10.1101/2025.03.14.643370)) or through transplantation (zebrafish, e.g., Barsh et al. 2017). We again nod to the technical infeasibility of such an “ideal” experiment in our system until such genetic characterizations exist for vestibular ganglion subtypes. In lieu of these experiments, we’ve expanded the Discussion (lines 392-400) to consider the necessary groundwork that future work must lay for our field to progress beyond indirect relationships between birth order, progressive spatial development, and functional circuit connectivity. We hope the Reviewer finds this addition adequately addresses their important concerns.

4) It should be noted that the idea that birth order controls central targeting of peripheral sensory neurons is not particularly new - e.g. was previously demonstrated for somatotopy of lateral line ganglion neurons specifically targeting distinct regions in the zebrafish hindbrain was previously demonstrated by Pujol-Marti and colleagues (doi: [10.1002/dvdy.22320](https://doi.org/10.1002/dvdy.22320); doi: [10.1523/JNEUROSCI.5157-11.2012](https://doi.org/10.1523/JNEUROSCI.5157-11.2012)).

We thank the Reviewer for catching our oversight in referencing the important prior work from Pujol-Marti et al, which used similar approaches (anatomical birthdating) to identify correlations between birth order, progressive spatially-restricted patterns of neurogenesis, and central axon outgrowth in a similar peripheral sensory system. We’ve taken care to update the Introduction (lines 48-50) and Discussion (lines 347) to include this important prior work, which along with the temporal axis drawn by Liu et al. 2022, motivates our examination of time as the “missing” variable for the vestibular sensory periphery, as for other peripheral sensory ganglia.

Reviewer 3:

Summary: This paper by Huang and colleagues examines the development of the statoacoustic ganglion and provides convincing evidence that the birth date and position of SAG cells correlates with their functional tuning and their connectivity to central and peripheral targets. Along with previous work from this group and others, this supports an elegant model in which birth order and topography likely shape the construction of an entire, evolutionarily conserved, sensorimotor pathway. The data and analyses are generally very good, with one exception, discussed below. A highlight of the paper was a clever experiment combining Kaede birthdating, “TIPM” calcium imaging, and laser-ablations that was used to show that sensory and central vestibular cells form functional circuits based on birth order and anatomical location. I expect the paper will certainly be of interest to the readership of Development.

We appreciate the Reviewer’s support for our manuscript and have implemented their thoughtful and detailed suggestions on how to improve the clarity and presentation of our conclusions. We acknowledge the shared consideration with Reviewer 2 as to the inferences of our birthdating experiments and have modified our conclusions accordingly in our revision.

Major comments:

1) Figure 1 makes the case for using a *hcrtr2* line to look at SAG neurons. The emphasis is on soma locations but this alone might not be strong evidence (for example if SAG neurons were intermingled with another cell type).

Electron microscopy reconstructions of the vestibular ganglion (Liu et al. 2022, DOI: <https://doi.org/10.1038/s41467-022-32824-w>), establish that SAG neurons are unlikely to be intermingled with other neuronal cell types. We've now referenced this foundational work in the text (line 75-78) to bolster our interpretation that the contiguous *hcrtr2*+ population contains anterior vestibular ganglion neurons. Additionally, we believe our new comparative analysis of *hcrtr2* with a published EM connectome of the vestibular ganglion (Fig. 1 and related Results) further supports our use of *hcrtr2* and more carefully considers and appreciates its limits.

2) Later in the paper, Kaede conversions show the *hcrtr2* cells project axons to the utricular macula and in the VIIIth nerve – why not include some of this anatomy in F1 to show the neurites go to the right places?

We thank the Reviewer for their suggestion and have expanded Figure 1 to include this anatomy.

3) My main concern with the paper related to figure 3 and the inferences about dorsoventral versus rostrocaudal developmental gradients. The data seems solid enough, but the results text (around Ln93 onwards) and logic of the argument is very difficult to follow. As far as I could understand the results, the authors observe early born cells in dorsal and caudal SAG (e.g. 22 and 26 hpf 'born before' cells in F3A). After that stage, cells seem to appear both rostrally and ventrally. Yet the authors somehow conclude that only DV development is of concern. ie "Dorsoventral, but not rostrocaudal development, is the primary predictor of late-born somatic accumulation..." and "Unlike its rostro-caudal functional organization, we discovered that the development of the vestibular ganglion proceeds along a dorsal-to-ventral axis." Perhaps a source of confusion relates to when the threshold is for describing cells as late-born?

We agree with the Reviewer that choosing the "right" timepoint to delineate "early" and "late" neurons is key, and we appreciate that our delineation here had an outsized impact on our interpretations: the caudal bias we observed was limited to 30% of *hcrtr2*+ neurons and constrained to <20% of total ganglion developmental time (8/50 hours). We've expanded on this in the Results (lines 136-139), as described above in response to Reviewer 2.

Overall, to address the shared concern between Reviewers 2 and 3 that we've insufficiently appreciated the early caudal development of the ganglion, we've taken great care to incorporate major new analyses (EM reconstructions and conversations with Tanimoto et al.), extensively discuss the limitations in variable transgenic access to the ganglion on our conclusions, and temper claims that imply dorsoventral organization is the only axis of importance. A more detailed explanation of these modifications are described above in response to Reviewer 2. We hope the Reviewer finds the revised manuscript is considerably stronger with this elaboration.

Whatever case, the data do clearly show there is a topography related to birth order in the SAG in which the earliest cells are localised dorso-caudally: Perhaps F6B might better have the pink cells confined to such a quadrant?

We've updated our model figure per the Reviewer's suggestion.

Minor comments:

1) Tanimoto et al 2022 is now published, Is there a reason to cite the biorXiv preprint (ref 24)? This reference has been updated; we thank the Reviewer for their attention to detail.

2) F1A: Inset also shows fluorescence micrograph (to visualise Kaede)?

This caption has been corrected to note that the image in F1A shows both transmitted and fluorescent light.

3) F1B: dashed circle is black. This has been corrected.

4) F1D: for the 220 neurons shown - presumably randomly selected?

F1D caption has been updated to reflect the n=220 neurons are a randomly selected subset.

5) In Table 1 and F1D its not very clear what is meant by "control". All transgenically labelled cells as an estimate of total SAG population? Similarly "The early-born cohort was caudal of center relative to control (pKS test=1.0x10⁻¹⁰). Mediolateral organization was not observed (pKS

test=0.25)": Again, not clear what is meant by control. Is the mediolateral KS test a comparison of the early vs late distributions, or something else?

We agree that it is important here to clarify that our "control" distributions refer to the n=345 neurons within the boundaries of the vestibular ganglion mapped from N=5 singly-expressing Tg(-6.7Tru.Hcrtr2:GAL4-VP16) larvae as originally shown in Figs. 1D and 1G-I. Accordingly:

- We've re-defined "control" in the Methods ("Statistical analysis of spatial organization"); as "reference dataset of n=345 neurons..."
- Updated appropriate table/figure captions (Table 1, F1D, F2D, F3A). In particular, Table 1 now lists "control" as a "Non-birthdated reference dataset", defined in the caption as the median number of neurons/fish from the n=345 reference dataset.
- Adjusted all references to "control" data in the relevant Results section ("Developmental time organizes...") to "**relative to our non-birthdated reference dataset**" with references to the n=345 neurons and Figs. 1D and 1G-I.

6) F3C-D y-axis need units **This has been corrected.**

7) At Ln99, the term "undifferentiated" neurons is introduced. What does this refer to?

We've changed this term to "unconverted" to clarify that these neurons were not yet born at the relevant conversion timepoint (line 175).

8) F4B: Dashed and dotted appear the same.

This has been corrected.

9) F4D-E: Suggest rewriting the legend text to state explicitly what is actually being imaged in D.

The cartoon suggests an axon tract en route to the macula yet the results text implies it is the macula.

We've adjusted the legend text for clarity: "**Vestibular ganglion projections terminating at the utricular macula....midpoint of vestibular ganglion projections**".

10) Ln136: "The matched connectivity..." Connectivity is not being assessed.

The Reviewer is correct. We've adjusted the subject of this line (line 219-220): "**Thus, early-born (dorsal) vestibular-ganglion neurons may innervate commonly-positioned (early-born, dorsal) central neuron partners.**"

11) In the key resources table and transgenic lines section of methods, please try to report the allele numbers for mutations and transgenes. I believe the table should list the genomic features for extant lines (of the form ZDB-ALT-...), not the construct identifier.

Genomic features and their identifiers are now reported in place of the construct identifier.

12) Ln313: **A measure of irradiance from the 405nm LED would make it easier to reproduce the photoconversions.**

We've performed this measurement (approx. 9 mW using a 9.5mm probe) and have included it in the Methods.

Second decision letter

MS ID#: dev.204616R1

MS TITLE: Birthdate aligns vestibular sensory neurons with central and motor partners across a sensorimotor reflex circuit for gaze stabilization

AUTHORS: Stephanie Huang, Emily Gershowitz, Marie R. Greaney, Samantha N. Davis, David Schoppik and Dena Goldblatt

Dear Dena,

I have now received two referees reports on the above manuscript, and have reached a decision. The referees' comments are appended below. The overall evaluation is positive and we would like to publish a revised manuscript in Development after you have considered the remaining concerns raised by one of the reviewers. Please attend to the reviewer's comments in your revised manuscript and detail them in your point-by-point response. If you do not agree with any of their criticisms or suggestions explain clearly why this is so.

Reviewer 1*Advance summary and potential significance to field*

I read all the responses of the authors to my comments and other reviewers comment, in addition to the revised manuscript. I am perfectly happy with the revised version of the manuscript and with all the answers, I especially loved the additional behavioral experiments that the authors choose to add. I recommend the publication of this exciting and carefully conducted study.

Reviewer 2*Advance summary and potential significance to field*

The authors have made substantive changes in response to my previous review. The additional analysis of the spatial distribution of neurons is a significant addition. In addition the authors have more fully addressed potential caveats to interpretations. They make a compelling case for temporal, "first-come, first-served" model. However the case for spatial organization remains much less compelling, and indeed the authors acknowledge so (e.g. lines 309-310), yet elsewhere a spatial component continues to be promoted (e.g. lines 7-8; 58-59; 311-312; 340-341). I think the authors should consider carefully how they use "spatiotemporal maps" when instead "temporal map" (e.g. line 377) would in many cases likely be sufficient.

Second revisionAuthor response to reviewers' comments

Reviewer 1: I read all the responses of the authors to my comments and other reviewers comment, in addition to the revised manuscript. I am perfectly happy with the revised version of the manuscript and with all the answers, I especially loved the additional behavioral experiments that the authors choose to add. I recommend the publication of this exciting and carefully conducted study.

We thank the Reviewer and appreciate their support for our manuscript's publication in Development.

Reviewer 2: The authors have made substantive changes in response to my previous review. The additional analysis of the spatial distribution of neurons is a significant addition. In addition the authors have more fully addressed potential caveats to interpretations. They make a compelling case for temporal, "first-come, first-served" model. However the case for spatial organization remains much less compelling, and indeed the authors acknowledge so (e.g. lines 309-310), yet elsewhere a spatial component continues to be promoted (e.g. lines 7-8; 58-59; 311-312; 340-341). I think the authors should consider carefully how they use "spatiotemporal maps" when instead "temporal map" (e.g. line 377) would in many cases likely be sufficient.

We are grateful for the Reviewer's support for our revised manuscript and have tempered language that implicates spatial maps (now "temporal maps", at their suggestion) as an organizing principle in all line numbers noted above. We note one exception: lines 311-312 (Discussion). We believe that "spatiotemporal" is appropriate here, as this Discussion sentence motivates the importance of future recoding experiments to determine the extent to which the maps we observe reflect the true progression of the ganglion ("Together, our considerations motivate future recordings in birthdated neurons to directly correlate the spatiotemporal patterns we observe with the anatomical and functional geometry of the vestibular ganglion.").

Third decision letter

MS ID#: dev.204616R2

MS TITLE: Birthdate aligns vestibular sensory neurons with central and motor partners across a sensorimotor reflex circuit for gaze stabilization

AUTHORS: Stephanie Huang, Emily Gershowitz, Marie R. Greaney, Samantha N. Davis, David Schoppik and Dena Goldblatt

Dear Dena,

I am happy to tell you that your manuscript has been accepted for publication in Development, pending our standard publication integrity checks.